# Interchromosomal interaction of homologous *Stat92E* alleles regulates transcriptional switch during stem-cell differentiation

Matthew Antel [1], Romir Raj[1], Madona Y. G. Masoud[1], Ziwei Pan[2,3], Sheng Li [2,3], Barbara G. Mellone[4,5] & Mayu Inaba [1✉]

Pairing of homologous chromosomes in somatic cells provides the opportunity of inter-chromosomal interaction between homologous gene regions. In the *Drosophila* male germline, the *Stat92E* gene is highly expressed in a germline stem cell (GSC) and gradually down-regulated during the differentiation. Here we show that the pairing of *Stat92E* is always tight in GSCs and immediately loosened in differentiating daughter cells, gonialblasts (GBs). Disturbance of *Stat92E* pairing by relocation of one locus to another chromosome or by knockdown of global pairing/anti-pairing factors both result in a failure of *Stat92E* down-regulation, suggesting that the pairing is required for the decline in transcription. Further-more, the *Stat92E* enhancer, but not its transcription, is required for the change in pairing state, indicating that pairing is not a consequence of transcriptional changes. Finally, we show that the change in *Stat92E* pairing is dependent on asymmetric histone inheritance during the asymmetric division of GSCs. Taken together, we propose that the changes in *Stat92E* pairing status is an intrinsically programmed mechanism for enabling prompt cell fate switch during the differentiation of stem cells.

[1] Department of Cell Biology, University of Connecticut Health Center, Farmington, CT, USA. [2] The Jackson Laboratory for Genomic Medicine, Farmington, CT, USA. [3] Department of Genetics and Genomic Sciences, University of Connecticut Health Center, Farmington, CT, USA. [4] Department of Molecular and Cell Biology, University of Connecticut, Storrs, CT, USA. [5] Institute for Systems Genomics, University of Connecticut, Storrs, CT, USA. ✉email: inaba@uchc.edu

D istant DNA regions interact not only in cis but also in trans to modify each other's gene expression states[1–3]. One fascinating facet of interchromosomal interaction occurs between homologous chromosomes in a phenomenon called homologous chromosome pairing. Although homologous chromosome pairing is most prominently studied in the context of meiosis, somatic cells of *Dipteran* including *Drosophila* also pair their homologs across the entire genome and throughout development[4–6]. Although the prevalence of complete pairing of homologs outside the germline in other organisms is still unclear, somatic pairing of specific chromosome regions does occur in a tightly regulated manner in many other organisms, including mammals (reviewed in[6]).

Haplotype-resolved Hi-C and/or fluorescent in situ hybridization (FISH) analyses have begun to reveal that homologous chromosome pairings have more global impact on 3D genome architecture and gene expression status than previously thought. For example, the strength of allelic pairing differs in a locus-specific and a tissue-specific manner[7] and correlates with local chromatin status[8]. This suggests that somatic homolog pairing may be under the control of a developmental program or extracellular signaling. However, the causal relationship between pairing and gene transcription is still uncertain.

How can the interaction between homologous chromosomes influence their transcriptional status? In flies, the consequence of somatic homolog pairing is represented by the phenomenon called transvection, whereby different mutant alleles of a gene-regulatory element can rescue each other's expression[9–14]. Transvection has been described for a number of *Drosophila* genes and can either promote or silence transcription (reviewed in[11]). Homolog pairing occurs between chromatin domains, called buttons, characterized by topology associated domains (TADs), which can be visualized by Hi-C[7], whereas transvection requires smaller DNA elements such as polycomb responsive elements (PREs) and insulator domains[7,11]. These requirements account for a recent report showing that pairing is necessary but not sufficient for transvection to occur[7]. Even though the phenomenon of transvection was first reported over 60 years ago[15], its mechanism is still not fully understood. Moreover, because the majority of transvection studies are transgene-based, whether endogenous wild type genes require trans-homolog interactions to modify their expression level is unclear.

*Drosophila* male germline stem cells (GSCs) constantly undergo asymmetric cell division (ACD) to produce one GSC and one gonialblast (GB) (Fig. 1a). The GB initiates the differentiation program to enter 4 rounds of syncytial divisions where the GB differentiates into spermatogonia (SG) that then become spermatocytes (SCs) (Fig. 1a). SCs then enter two rounds of meiotic divisions and ultimately differentiate into functional sperm. Upon exit from the GSC state when they are born, GB's must downregulate key stem cell-specific genes and must switch on genes required for differentiation. Several studies have investigated the extrinsic signals and intrinsic factors required for proper cell fate transition during the asymmetric division of GSCs (reviewed in[16]). In particular, major niche ligands, Unpaired (Upd) and Decapentaplegic (Dpp), have been believed to be factors in dictating stem cell fate, as ectopic expression of Upd or Dpp induces overproliferation of GSC-like cells outside of the niche[17,18]. On the other hand, a number of studies have identified the need for intrinsic factors (reviewed in[19]). For example, there is a biased segregation of histones H3 and H4 in GSCs and GBs. Newly synthesized histone H3 is preferentially incorporated on sister chromatids that are inherited by the GB during asymmetric division, while the old histone H3 remains in the GSC[20]. Perturbation of this asymmetric histone H3 inheritance results in differentiation defects[21], demonstrating the indispensability of cell-intrinsic mechanisms. However, it is unclear how such mechanisms collaborate with each other to successfully produce cells with distinct cell fates. Moreover, the ON/OFF timing of key regulatory genes during the fate transition have yet to be elucidated.

In this study, we investigate the change of pairing state of a stem cell-specific gene, *Stat92E*, and provide the potential role of local pairing on transcriptional downregulation during the process of stem-cell differentiation.

## Results

**The *Stat92E* locus shows a distinct pairing pattern after ACD.** Signal transducer and activator of transcription 92E (Stat92E) is a direct downstream molecule of the niche signal Unpaired (Upd) and is known to be required for GSC establishment and maintenance[17,22–25]. Stat92E protein is specifically expressed in the GSC population and decreases immediately in differentiating daughter cells, called gonialblasts (GBs)[26]. To determine whether *Stat92E* expression is transcriptionally regulated, we designed single molecule fluorescent in situ hybridization (FISH) (smFISH)[27–32] probes targeting an exon or an intron of the *Stat92E* gene to monitor the levels of *Stat92E* transcription (Fig. 1b). The intron probe hybridizes to *Stat92E* nascent transcript, which represents active transcription occurring on the template DNA region[29,33,34] (seen as puncta double positive for exon and intron probe signal, whereas mature mRNA shows only exon probe signal, Fig.1b). Specificity for both probes was validated by using *Stat92E* null clones (Supplementary Fig. 1a, b).

Unexpectedly, upon visualizing *Stat92E* nascent transcripts, we noticed that the localization pattern of the *Stat92E* locus is different between GSCs and differentiating cells. In GSCs, we tended to observe a single focus of nascent transcript with high fluorescence intensity. In GBs and SGs, we tended to observe two separate puncta (Fig. 1c, d). This suggested the possibility that the *Stat92E* locus on homologous chromosomes may be paired in stem cells, but become unpaired upon differentiation. The observed changes in the pairing pattern of the *Stat92E* locus prompted us to investigate whether such pairing is under the control of early germline development and if it has any impact on the *Stat92E* transcript.

It should be noted that a pair of GSC and GB enters S-phase immediately after mitosis when cells are still interconnected (Fig. 1e) and cytokinesis occurs after cells enter G2 phase[35]. S-phase cells sometimes have multiple (3 or more) foci of *Stat92E* intron FISH signal (Fig. 1f), likely reflecting the separation of sister chromatids during DNA synthesis. Therefore, we excluded cells with more than three foci from the following pairing assays.

**_Stat92E_ expression is transcriptionally regulated during differentiation.** Because the mammalian STAT3 (a homolog of Stat92E) protein is known to bind to its own promoter to induce its own expression[36], it has been believed that the GSC-specific *Stat92E* expression pattern in the *Drosophila* testis might be similarly regulated by a transcriptional feedback loop[37]. However, this model has never been tested in the GSC niche.

To test whether *Stat92E* expression is regulated at the transcriptional level, we quantified *Stat92E* mRNA levels using the *Stat92E* exon probe. smFISH visualizes single molecules of *Stat92E* mRNA as uniform sizes and intensities of dots in the cytoplasm of developing germ cells and surrounding other cell types, including hub cells, somatic cyst stem cells (CySCs) and cyst cells (CCs) (Fig. 1g). To quantify mRNA levels in early stages of germ cells, we manually counted the number of smFISH dots. Since germ cells and somatic CySCs or CCs, which are both positive for *Stat92E* mRNA, closely adhere to each other, smFISH signals in germ cells and in somatic cells overlap at the cells' surfaces (Fig. 1g

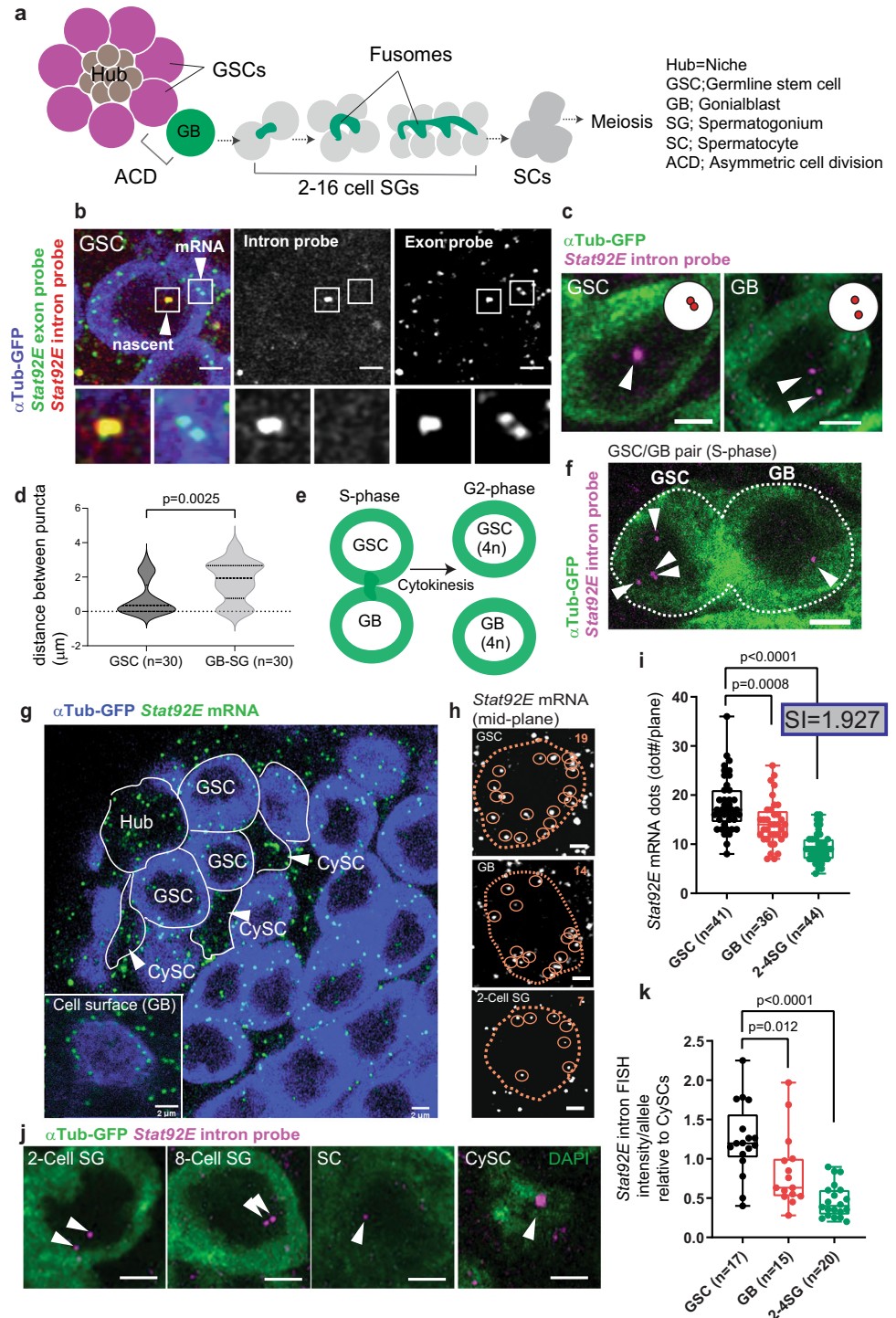

inset). Therefore, we counted the smFISH dots in a single confocal plane at approximately the middle section of germ cells (Fig. 1h, Supplementary Fig. 1 for more details). We found that *Stat92E* mRNA downregulation occurs gradually as differentiation proceeds from the GSC stage to the SG stage (Fig. 1g–i). We defined the ratio of *Stat92E* mRNA levels between GSC and 2–4 SG stages (GSC/2-4SG), as the silencing index (SI). We found that in wild type testes, the SI was close to 2 (1.927), meaning a nearly 2-fold decrease in mRNA level as differentiation proceeds.

Although the smFISH results provided precise quantification of *Stat92E* mRNA, the mRNA can be diluted out upon cell division, leaving open the possibility that the change of transcription occurs

even earlier in differentiation. Quantification of intron signal intensity relative to that in CySCs, which show consistently high signal in a single focus, also indicated that the transcription of *Stat92E* decreases gradually as differentiation proceeds, (Fig. 1j, k). Both results indicate that *Stat92E* transcription is not completely shut off at the GB stage but is gradually downregulated during differentiation.

## The change in *Stat92E* pairing states is locus- and cell type-specific.

To confirm the pairing state of the *Stat92E* locus at the DNA level, we performed OligoPaint DNA FISH[38] on the *Stat92E*

**Fig. 1 The Stat92E locus shows a distinct pairing pattern after ACD. a** A schematic of the early germline differentiation in *Drosophila* testis. **b** A representative image of *Stat92E* nascent transcript and mRNA in GSC. Lower panels show magnified regions from insets in the top images. Nascent transcripts are detected by both an intron probe (red) and an exon probe (green) in the nucleus. **c** Representative images of the nascent transcript of *Stat92E* (magenta, white arrowheads) in GSC (Left) and GB (Right). **d** Violin plots show distances between *Stat92E* nascent transcript puncta in GSCs vs. GB-SGs. The *p* value was calculated with two-tailed Student's *t*-test. **e** Estimated cell cycle stage of a GSC and GB pair by cell connectivity. **f**) A representative image of a GSC-GB pair in S-phase. White arrowheads indicate multiple puncta of nascent transcript (magenta). **g** A testis-tip image of *Stat92E* mRNA FISH. GSCs and CySCs are marked by white lines. **h** Examples of mRNA quantification from mid-plane of cells in indicated stages. mRNA molecules (smFISH dots) are encircled by solid-line circles. **i** Quantification of *Stat92E* mRNA (smFISH dots/plane) at the indicated stages. SIs (see texts) are shown in boxes above bars. **j** Representative images of *Stat92E* nascent transcript (magenta, white arrowheads) in the cells at the indicated stages. **k** Quantification of the intensity of *Stat92E* nascent transcript. Y axis indicates fluorescence intensity of nascent transcript signal in indicated stage of germ cells divided by CySCs' (see Method for details). To plot intensity/single allele, measured intensities of paired cases were divided by two. Note that the data in all stages contain fractions with extremely low nascent transcript level, likely reflecting S-phase cells in which transcription is silent. In all RNA FISH experiments, *nos* > αTubulin-GFP flies were used to identify stages of germline (blue in **b** and **g**, green in **c**, **f** and **j**). Scale bars represent 2µm. Box plots show 25–75% (box), median (band inside) and minimum to maximum (whiskers) with all data points. The adjusted *p* values were calculated by one-way anova with Dunnett's multiple comparisons for comparing each dataset with GSC data in (**i**) and (**k**). All plotted data points are provided in Source Data.

locus, which is located on the right arm of chromosome 3. A previous study showed that homolog pairing occurs between button regions in homologous chromosomes, often containing a full TAD[7]. Therefore, we selected a 60Kb OligoPaint probe target region spanning the entire *Stat92E* gene region within a TAD (estimated based on a published Hi-C sequence data analysis, Fig. 2a). Similar to the pattern observed in our intron FISH experiment (Fig. 1c, d), the *Stat92E* locus tended to be paired in GSCs, detected as a single focus in a nucleus, and was often unpaired, detected as two foci in a nucleus, in differentiating GBs (Fig. 2b, c) (see[39–42] and Supplementary Fig. 2 for the method used to estimate 3D-distance and Supplementary Fig. 3a, b for validation of the probe specificity using an antisense probe).

Unlike somatic cells, homologous chromosomes are unpaired in *Drosophila* female GSCs and become paired as differentiation proceeds in order to prepare for meiosis[43]. In contrast, it has been reported that the male germline shows paired patterns of satellite DNA regions throughout differentiation[44], suggesting that the observed pairing/unpairing events could be unique to the *Stat92E* locus. To test whether *Stat92E* pairing is locus-specifically regulated, we performed DNA FISH on flies carrying an array of *lacO* repeats inserted at an euchromatic region close to where *Stat92E* is located on chromosome 3 (Fig. 2d–f) or at a heterochromatic region near the telomere on chromosome 2 (Fig. 2g–i) using oligonucleotide probes targeting the *lacO* repeat sequence. We observed that the *lacO* locus on both regions remained paired throughout differentiation (Fig. 2d–i), suggesting that the observed pairing/unpairing event seen within the *Stat92E* region occurs in a locus-specific manner.

The distance between *Stat92E* puncta progressively decreased as SGs differentiated into SCs (Fig. 2c), presumably in preparation for meiotic pairing pattern which takes place in the SC stage[45]. Consistent with previous reports, early SCs show a uniformly paired pattern in all genotypes (Supplementary Fig. 3c)[45]. Similar to early SCs, the somatic cyst stem cells (CySCs) constantly showed a paired pattern at the *Stat92E* locus (Supplementary Fig. 3d, e). These results suggest that the observed paired/unpaired change is cell type- and stage-specific.

A previous study demonstrated that local pairing can occur independently of entire homolog pairing even when alleles are distantly positioned[7]. We tested if *Stat92E* pairing is dependent on its position within homologous chromosomes. In flies carrying the TM3 balancer chromosome (FlyBase ID: FBba0000047) in which the *Stat92E* locus is inverted and dislocated approximately 10 Mb away from its original location[46,47], we still observed a similar pattern of *Stat92E* pairing/unpairing (Fig. 2j–l), suggesting that the *Stat92E* pairing regulation still occurs between alleles even if they are dislocated.

The observed unpairing event occurs between two homologous chromosomes and not between sister chromatids, as we observed only one dot of DNA FISH signal in files heterozygous for deletion of the *Stat92E* locus (*Df(3 R)BSC516*, Fig. 2m–o). This result also indicates that the *Stat92E* OligoPaint probe was specifically recognizing the *Stat92E* locus.

**The change in Stat92E pairing states does not reflect the difference in cell-cycle stages.** It is thought that somatic homolog pairing may be regulated cell cycle dependently[5,48]. Therefore, we wondered if the observed *Stat92E* pairing change during differentiation is developmentally regulated or rather simply reflects the changing fractions of cells in different cell cycle stages. To distinguish these possibilities, we examined the pairing states of *Stat92E* between GSCs and differentiating cells when both populations were in the same cell cycle stage. To this end, we expressed Fly-Fucci, a fluorescent cell cycle marker[49], under the control of the germline driver *nosGal4*, and examined *Stat92E* pairing states by visualizing nascent transcript foci using intron RNA FISH (Fig. 3a). A previous study has demonstrated that early germ cells in the *Drosophila* testis lack G1 phase and most germ cells are in G2 or S-phase[35]. We noticed that *Stat92E* transcription became low, often undetectable during S-phase (Fig. 3a). Therefore, we determined the distance between homologous *Stat92E* loci in G2-phase cells. The distribution of distances between *Stat92E* puncta measured specifically in G2-phase cells did not have a significant difference from the measurement of entire cell populations (Fig. 3b, c), indicating that the observed change of frequency of *Stat92E* pairing between the GSC and the GB stage was unlikely caused by the difference of cell cycle stages.

**The change in Stat92E pairing states is required for subsequent silencing of transcription.** The locus-specific pairing state change we observed from GSC to GB prior to the downregulation of *Stat92E* transcription led us hypothesize that the pairing change may be required for subsequent *Stat92E* downregulation. To test this hypothesis, we determined the timing of *Stat92E* downregulation during the early stages of germline development in several genotypes. To compare *Stat92E* downregulation across different genotypes, we used the silencing index (SI) as defined above.

Flies heterozygous for the deficiency of the *Stat92E* locus (*Df(3 R)BSC516*, which lacks 3 R: 20,093,311..20,790,571 and therefore lacks one copy of the entire TAD including *Stat92E* (3 R: 20,470,000..20,570,000), Fig. 2m) should lack the effect of trans-chromosomal interaction of the *Stat92E* locus. Indeed,

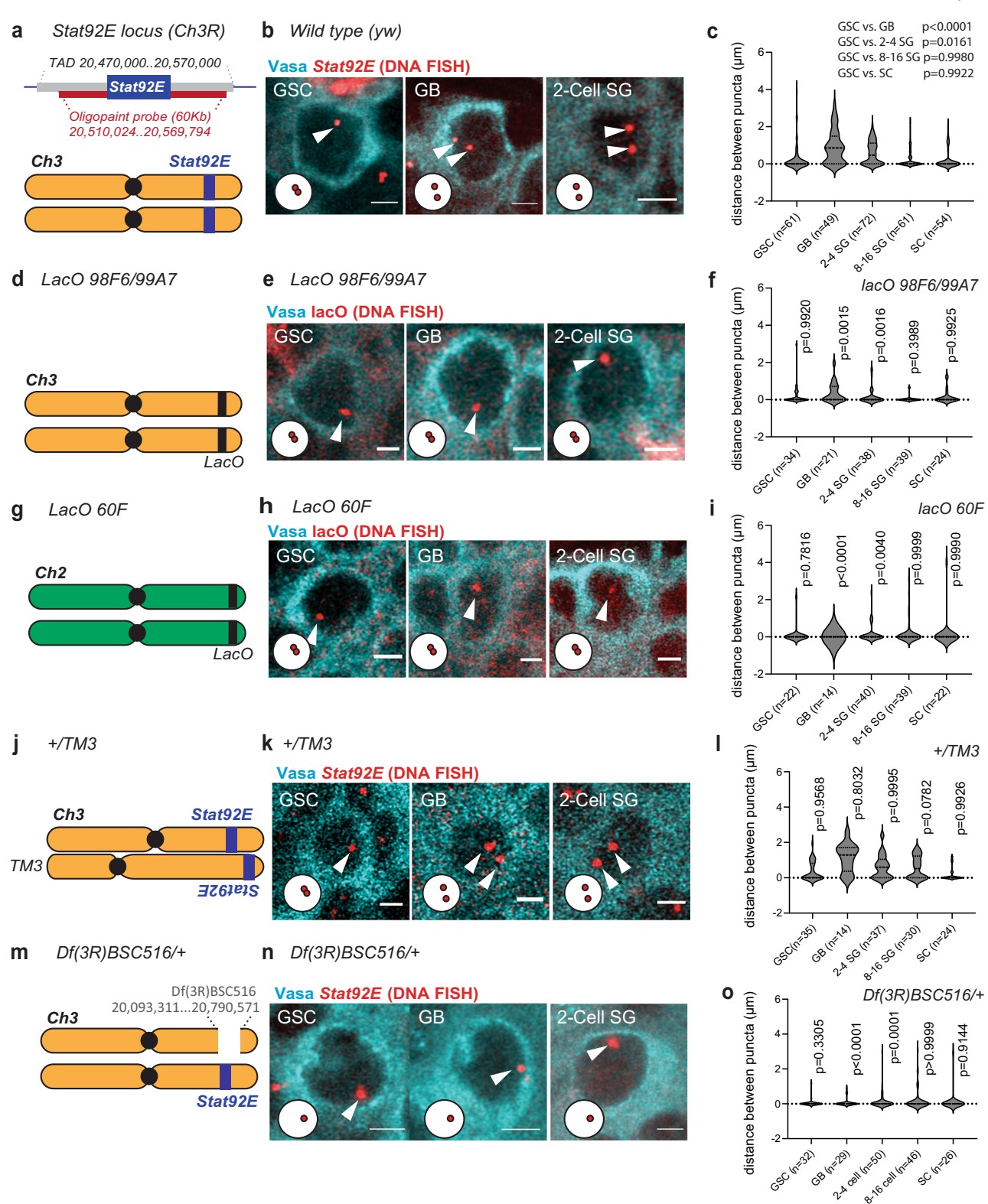

**a** *Stat92E locus (Ch3R)*

**b** *Wild type (yw)*

**c** *yw*

**d** *LacO 98F6/99A7*

**e** *LacO 98F6/99A7*

**f** *lacO 98F6/99A7*

**g** *LacO 60F*

**h** *LacO 60F*

**i** *lacO 60F*

**j** *+/TM3*

**k** *+/TM3*

**l** *+/TM3*

**m** *Df(3R)BSC516/+*

**n** *Df(3R)BSC516/+*

**o** *Df(3R)BSC516/+*

we found that *Stat92E* mRNA level failed to decrease in *Df(3 R) BSC516+* testes as differentiation proceeds. While in control testes the SI was close to 2 (1.898), meaning a nearly 2-fold decrease in mRNA level as differentiation proceeds, the SI in *Df(3 R)BSC516+* testes was close to 1 (1.237) (Supplementary Fig. 1h), indicating that the mRNA level remained similar throughout GSC to 2-4 SG stages. Nascent transcript intensity

of *Df(3 R)BSC516+* was also unchanged (Supplementary Fig. 4), indicating that the change of mRNA level reflects a change of transcriptional activity rather than mRNA stability.

Since *Df(3 R)BSC516/+* lacks one copy of the *Stat92E* gene, in addition to the defective SI, it also showed lower *Stat92E* mRNA expression throughout all stages (Supplementary Fig. 1f, h) presumably due to having only one copy of the gene. This made it

**Fig. 2 The change in *Stat92E* pairing states is locus- and cell type-specific.** Left columns (**a**, **d**, **g**, **j**, **m**); A schematic of the *Stat92E* locus on chromosome3 (**a**). Estimated TAD boundaries and the region recognized by the *Stat92E* OligoPaint DNA FISH probe sets are shown at the top (**a**). A schematic of the *lacO 98F6/99A7* insertion (**d**), the *lacO 60 F* insertion (**g**), the position of the *Stat92E* locus on chromosome 3 and on the TM3 balancer (**j**), and the *Stat92E* deficiency, *Df(3 R)BSC516* (**m**). Middle columns (**b**, **e**, **h**, **k**, **n**); **b**, **k**, **n** Representative images of DNA FISH targeting the *Stat92E* locus in the indicated stages of germ cell development in the indicated genotypes; (**b**) wild type (yw), (**k**) heterozygous for TM3 (+/TM3), (**n**) heterozygous for *Stat92E* deficient allele (*Df(3 R)BSC516/*+), **e**, **h** Representative images of DNA FISH targeting *lacO* locus in the indicated stages of germ cells development in *lacO 98F6/99A7* homozygous (**e**) or *lacO 60 F* homozygous (**h**) genotypes. In all DNA FISH samples, germ cells were visualized by Vasa staining (cyan). DNA FISH signals are shown in red (pointed by white arrowheads). Representative pairing states are shown in lower left corner of each image. All scale bars represent 2 μm. Violin plots (right columns, **c**, **f**, **l**, **l**, **o**); Violin plots show the distance between puncta of DNA FISH corresponding to the experiment shown in the middle panels (see details for Supplementary Fig. 2). Although most of the cells in *Df(3 R)BSC516/*+ flies showed only single spot plotted as distance zero (**o**), we also detected cells which had 2 puncta within a single nucleus in a low frequency (~10%), likely representing separated sister chromatids. Violin plots show KDE (kernel density estimate) and quantile lines and the width of each curve corresponds with the frequency of data points. The adjusted *p* values are calculated using Dunnett's multiple comparisons comparing with GSC data for **c**. For other graphs, *p*-values were calculated by comparing with data shown in **c** using Šidák's multiple comparisons. All plotted data points are provided in Source Data. Number of scored cells, which are randomly chosen from at least 10 testes for each experiments, is shown for each data point.

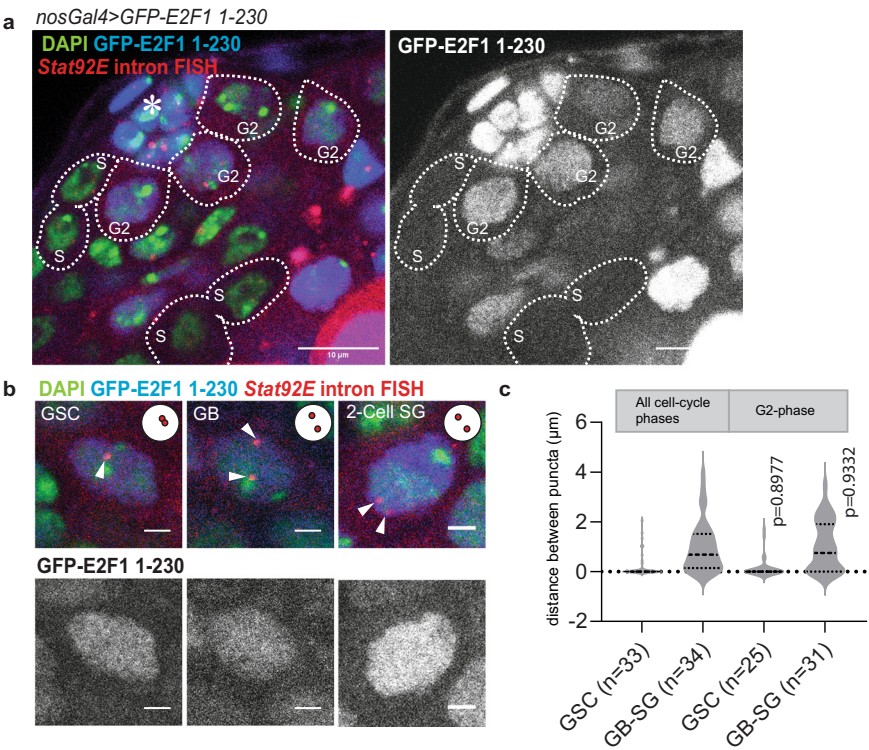

**Fig. 3 The change in *Stat92E* pairing states does not reflect the difference in cell-cycle stages. a** A representative image of *Stat92E* nascent transcript (red) in Fly-Fucci testis. GFP-E2F1-230 (blue) is positive in G2-phase and negative in S-phase cells. White dotted lines encircle germ cells in indicated cell cycle stages. DAPI is shown in green. Right panel shows GFP channel. *Stat92E* nascent transcript was typically weak or undetectable in S-phase cells. **b** Representative images of G2 phase germ cells in indicated stages. *Stat92E* intron FISH signal (white arrowheads) indicate pairing states at the indicated stage of germ cells. Representative pairing states are shown in the upper right corner of each image. Lower panels show GFP channel. **c** Violin plots show distances between *Stat92E* nascent transcript puncta at the indicated stages of germ cell development measured in all cell-cycle stages or only in G2-phase cells. Violin plots show KDE and quantile lines and the width of each curve corresponds with the frequency of data points. The adjusted *p* values were calculated with Šidák's multiple comparisons. All plotted data points are provided in Source Data. Number of scored cells, which are randomly chosen from at least 10 testes for each experiments, is shown for each data point. Scale bars in (**a**) represent 10 μm. Scale bars in b represent 2 μm.

difficult to judge the downregulation timing. Therefore, we attempted to identify conditions in which two copies of functional *Stat92E* locus are still present but they do not pair. Hence, we examined if the endogenous *Stat92E* locus can pair with a *Stat92E* transgene located on another chromosome. To test this, we introduced a bacterial artificial chromosome (*BacTg*) harboring the entire *Stat92E* locus within a 80Kb region (*Bac VK00037*) on chromosome 2 into the *Stat92E* deficiency background, *Df(3 R)BSC516* (*BacTg/Df* for short; Fig. 4a). In DNA FISH experiments examining the *BacTg/Df* testes, we consistently

observed two spots in all stages, representing a failure of the endogenous *Stat92E* gene on chromosome 3 to pair with the *BacTg* on chromosome 2 (Fig. 4b, c). A previous study demonstrated that transgenes can pair even when they are separately positioned when the transgenes contain the full TAD[7]. The *Stat92E* Bac construct (*Bac VK00037*) lacks ~65Kb of the proximal portion from the predicted TAD, suggesting the requirement of this region for pairing.

Taking advantage of the *BacTg/Df* line, in which there are two functional copies of the *Stat92E* gene but they never pair, we next

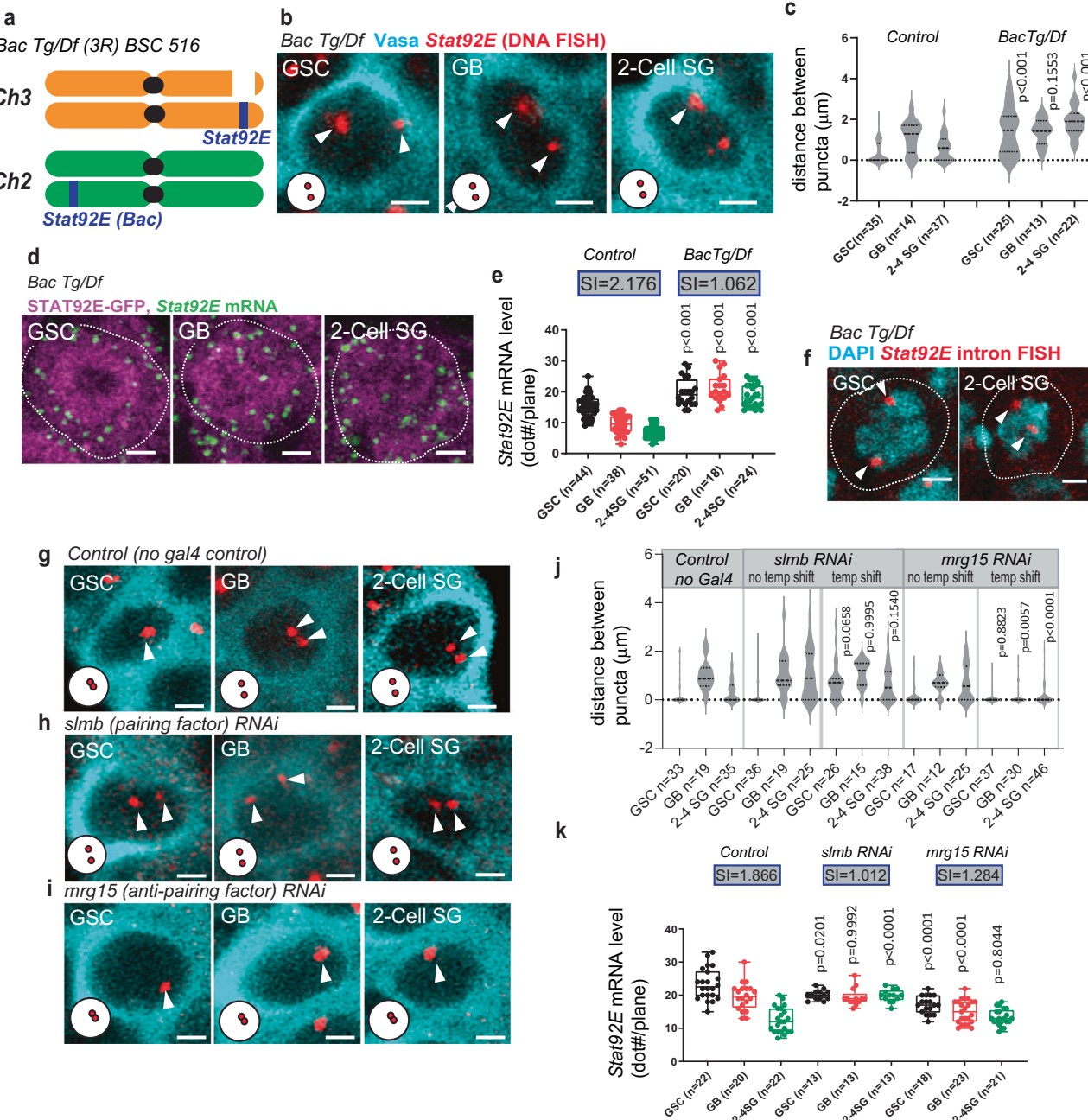

**Fig. 4 The change in *Stat92E* pairing states is required for subsequent silencing of transcription. a** A schematic of *BacTg/Df(3 R)BSC516* (*BacTg/Df*) genotype. **b** Representative images of *Stat92E* DNA FISH puncta (red, white arrowheads) at the indicated stages of germ cells (Vasa, cyan) in *BacTg/Df* genotype. **c** Violin plots show distances between DNA FISH puncta. **d** Representative images of *Stat92E* mRNA (exon probe smFISH, green) at the indicated stages of germ cells in indicated genotypes. **e** Quantification of *Stat92E* mRNA levels. *Y* axis values are the number of mRNA dots present in a middle plane of the cell (dot#/plane). SIs (see text) are shown in boxes above bars. **f** Representative images of *Stat92E* nascent transcript (intron probe, red) at the indicated stages of germ cells in *BacTg/Df* testes. DAPI (cyan). **g**, **h**, **i** Representative images of *Stat92E* DNA FISH (red, white arrowheads) at the indicated stages of germ cells (Vasa, cyan) in indicated genotypes. **j** Violin plots show distances between DNA FISH puncta. **k** Quantification of *Stat92E* mRNA levels. Y axis values are the number of mRNA dots scored in a middle plane of the cell (dot#/plane). SIs (see text) are shown in boxes above bars. Adjusted *p* values are provided for comparison between control (wild type, yw) and *BacTg/Df* data in (**c**) and (**e**) and comparison between temperature-shifted experiment (temp shift) and no temperature-shift control (no temp shift) in (**j**) and (**k**). The adjusted *p* values were calculated using Šidák's multiple comparisons. Violin plots show KDE and quantile lines and the width of each curve corresponds with the frequency of data points. Box plots show 25–75% (box), median (band inside) and minimum to maximum (whiskers) with all data points. All plotted data points are provided in Source Data. Number of scored cells, which are randomly chosen from at least 10 testes for each experiments, is shown for each data point. For RNAi experiments, temperature sensitive *nosGal4* driver, *nosGal4^ts^*, was used. All scale bars represent 2 μm. Representative pairing states are shown in lower left corner of each image.

investigated if the regulation of *Stat92E* transcription is compromised in this genotype. We examined the change of mRNA levels during differentiation in *BacTg/Df* testes and found that germ cells failed to downregulate *Stat92E* during differentiation, with a mRNA SI of nearly 1 (1.062, Fig. 4d, e) and a nascent transcript SI of 0.810 (Supplementary Fig. 4).

It is possible that the Bac transgene placed in a different location may be subjected to different positional effects that impact the gene downregulation timing. To exclude this possibility, we compared *Stat92E* nascent transcript levels between the endogenous locus and the *BacTg* locus. We did not detect noticeable differences of nascent transcript intensity between the two puncta (Fig. 4f), indicating that the *BacTg* and the endogenous *Stat92E* locus are expressed at similar levels. Therefore, the observed defect of *Stat92E* downregulation in *BacTg/Df* is unlikely due to a position effect of the *BacTg* allele and is most likely due to the absence of pairing between *BacTg* and the endogenous *Stat92E* locus.

Next, we tested the effect of global pairing and anti-pairing factors on *Stat92E* downregulation. Condensin II, a DNA loop extrusion factor, has been shown to antagonize homolog pairing[48,50,51], and its interacting factor, the *Drosophila* homolog of human MORF4-related gene on chromosome 15 (Mrg15), is known to be an anti-pairing factor[52]. The Condensin II complex is inactivated when its subunit Cap-H2 is degraded by the SCF (Skp/Cullin/F-box) E3 ubiquitin-ligase-Slimb complex. Therefore, the component Supernumerary limbs (Slmb) functions as a pairing promoting factor[50,53]. We observed that knockdown of *Slmb* in the germline under the *nosGal4* driver significantly decreased *Stat92E* pairing in GSCs, consistent with Slmb's role as a pairing factor (Fig. 4g, h, j). In contrast, knockdown of the anti-pairing factor *Mrg15* significantly increased *Stat92E* paring in GB-SGs (Fig. 4g, i, j). In both conditions, the change in pairing state between GSC and GB-SG did not occur and *Stat92E* failed to be downregulated during stage progression (Fig. 4k), with each genotype having a SI close to 1 (1.012 for *nos(ts) > Slmb RNAi*, and 1.284 for *nos(ts) > Mrg15 RNAi*). These results strongly suggest that the change in pairing state from the GSC to GB stage promotes prompt downregulation of *Stat92E* transcription.

**Stat92E nascent transcript levels differ in paired and unpaired conditions.** To assess the direct effect of pairing change on downregulation of transcription, we next attempted to examine levels of nascent transcript between paired and unpaired fractions of cells within the same stage. We specifically focused on the GB stage, when the unpaired populations are undergoing the switch from a paired state. The pair of interconnected GSC and GB has been known to synchronously enter S-phase when they remain continuous prior to cytokinesis (Fig. 5a)[35]. At this phase, *Stat92E* nascent transcript was almost undetectable (Fig. 3a). After completion of cytokinesis, GSCs and GBs are in G2 phase and contain already duplicated chromosomes (4n, Fig. 5a). To avoid effects of the cell cycle on transcription, we selected G2-phase GBs, identified as cells located one cell layer away from the hub and no longer connected to GSCs, and compared their levels of nascent transcript between paired and unpaired alleles.

If each *Stat92E* allele contains the same level of nascent transcript, the ratio of intron RNA FISH intensity between a paired and unpaired pattern is expected to be 2:1. However, we found that the average intensity of intron signal per allele was approximately 60% of the expected level (the ratio of paired vs. unpaired loci was approximately 2:0.6, Fig. 5b, c). In contrast, OligoPaint DNA FISH intensity between paired and unpaired fractions showed a ratio of 2:1 as expected (Fig. 5d, e). These results support our hypothesis that physical interaction of

homologous regions impacts the levels of nascent transcript, either enhancing transcript between paired alleles and/or suppressing transcript upon unpairing (Fig. 5f).

**Regulation of pairing requires the Stat92E enhancer but not transcription.** To determine the requirement of *cis* regulatory elements of *Stat92E* for pairing, we examined a *Stat92E* mutant allele, *Stat92E[06346][54]*, in which a p-element is inserted into a putative *Stat92E* intronic enhancer (Fig. 6a). In *Stat92E[06346]/+* heterozygous animals, nascent *Stat92E* was only detectable on a single locus, while DNA FISH shows two discrete spots in GB and SG, indicating that, as expected, the *Stat92E[06346]* allele completely lacks transcription as reported previously[54] (Fig. 6b–d). In *Stat92E[06346]/+* flies, we found that pairing at the *Stat92E* region was completely disrupted in GSCs as compared to wild type (Fig. 6e, f), suggesting that the *Stat92E* enhancer element is necessary for proper pairing in GSCs. The reduced expression of *Stat92E* mRNA in all germ cells was maintained at a similar level throughout differentiation (Fig. 6g, h) with a SI near 1 (1.172), consistent with our model that a change in pairing state is required for prompt downregulation of *Stat92E*.

Next, we asked whether or not the effect of *Stat92E[06346]* on pairing is caused by the lack of transcriptional activity from the allele. We attempted to artificially activate transcription of endogenous *Stat92E* allele in cis using the flySAM technique[55], which induces transcription by combining sgRNA targeting the *Stat92E* transcription start site (3 R:20,552,774..20,552,796 [+]) with dCas9 fused to VPR (VP64-p65-Rta), a tripartite transcriptional activator domain (Fig. 6i). Compared to the control, germline driver *nosGal4*-induced flySAM caused increased *Stat92E* expression levels throughout differentiation (Fig. 6j, o). However, the *Stat92E* pairing was unchanged (Fig. 6k, p), suggesting that pairing regulation is unlikely to be downstream of transcriptional activity.

To confirm the effect of *Stat92E* transcription on pairing, we next blocked transcription in cis at the *Stat92E* locus using the dCas9-mediated transcriptional knockdown, CRISPRi, combining dCas9 overexpression with sgRNA targeting the transcription start site, Fig. 6l)[56]. As with the flySAM results, even though CRISPRi caused a decrease in mRNA expression (Fig. 6m, o), it did not affect the pattern of pairing (Fig. 6n, p). These data suggest that the observed change in pairing state at the *Stat92E* region is not a consequence of a change in transcriptional activity in cis.

**Stat92E pairing is under the control of asymmetric histone inheritance.** During the asymmetric division of the GSC, a sister chromatid incorporated with parental histone H3 and H4 is preferentially retained in the GSC, while the other sister chromatid that incorporates newly synthesized histone H3 and H4 is inherited by the GB[20]. This pattern has been proposed to influence distinct chromatin states between GSCs and GBs. Notably, the change we observe in *Stat92E* pairing state was already apparent in GBs, immediately after asymmetric division. Therefore, we tested whether or not asymmetric histone inheritance contributes to the distinct pairing states of the *Stat92E* locus between GSC and GB. To this end, we perturbed the histone H3 inheritance asymmetry by expressing a mutant form of histone H3 that cannot be phosphorylated at Thr3 (H3T3A), resulting in the random distribution of pre-existing and newly synthesized histone H3 between GSC and GB[21]. Expression of histone H3T3A-GFP in the germline using the *nosGal4* driver resulted in the *Stat92E* locus remaining mostly paired throughout differentiation, while the control testes expressing wild type histone H3-GFP showed the expected pairing/unpairing switch seen in

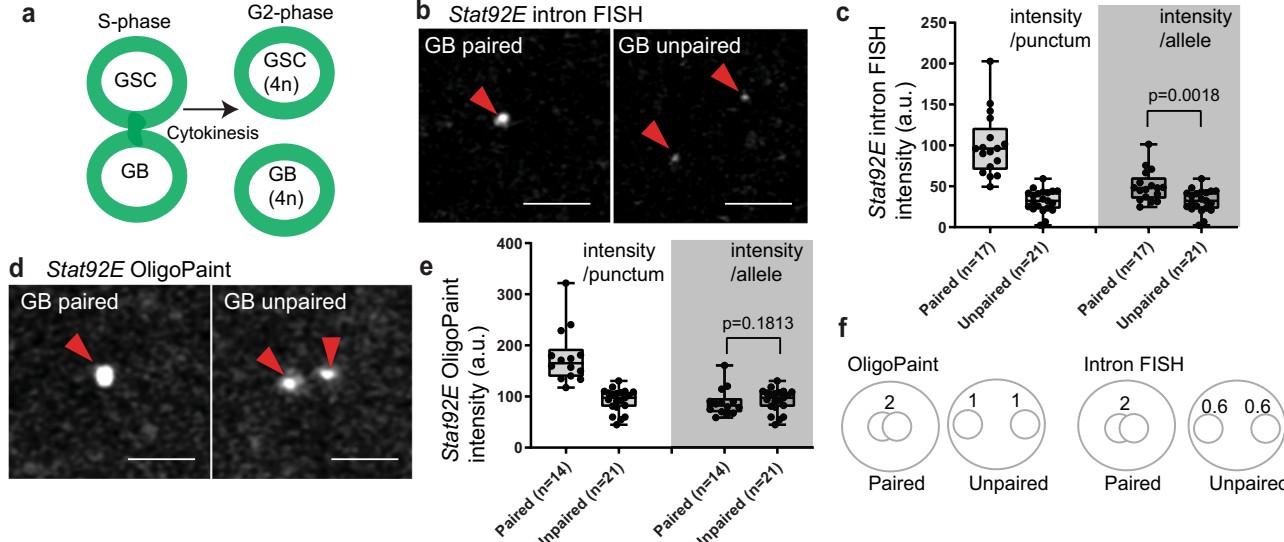

**Fig. 5 *Stat92E* nascent transcript level differs in paired and unpaired conditions. a** Schematic of cell cycle stages of a GSC and GB pair. **b** Representative images of *Stat92E* nascent transcripts in GBs. Left and right panel show paired or unpaired *Stat92E* homologous alleles, respectively. Red arrowheads indicate *Stat92E* nascent transcripts. **c** Quantification of nascent transcript intensity (measured as a ratio relative to nascent transcript intensity of CySC's, see Method for details) in paired and unpaired conditions of GBs. Left two columns show intensity/punctum. Right two columns show intensity/allele. To plot intensity/allele, measured intensities of paired cases were divided by two. **d** Representative images of *Stat92E* DNA FISH in GBs. Left and right panel show paired or unpaired *Stat92E* homologous alleles, respectively. Red arrowheads indicate *Stat92E* locus visualized by DNA FISH. **e** Quantification of DNA FISH signal intensity in paired and unpaired conditions of GBs. Left two columns show intensity/punctum. Right two columns show intensity/allele. To plot intensity/allele, measured intensities of paired cases were divided by two. **f** A schematic of the intensity measurement results shown in (**c**) and (**e**). Box plots show 25–75% (box), median (band inside) and minimum to maximum (whiskers) with all data points. The *p* values were calculated by two-tailed Student's t-tests. All scale bars represent 2 µm. Number of scored cells, which are randomly chosen from at least 10 testes for each experiments, is shown for each data point.

wild type flies (Fig. 7a–c). *nos > H3T3A-GFP* did not affect the pairing states of a *lacO* insertion, which remained paired throughout differentiation (Fig. 7d), suggesting that asymmetric histone inheritance may have an effect specifically on the *Stat92E* locus. The difference in distance of DNA FISH puncta between *nos > H3-GFP* and *nos > H3T3A-GFP* was not due to compromised distribution of cell cycle stages in either genotype, as both showed a comparable frequency of 5-ethynyl-2'-deoxyuridine (EdU) incorporating S-phase cells (Supplementary Fig. 5a–c).

The failure to switch to an unpaired state in cells expressing histone H3T3A-GFP resulted in defective downregulation of *Stat92E* during differentiation (Fig. 7e–g), with a SI close to 1 (1.237), while the control cells expressing histone H3-GFP had a SI close to 2 (1.938), further supporting the idea that the change in pairing state of *Stat92E* is required for the downregulation of its expression.

To establish that the effect of histone H3T3A-GFP on pairing is specifically due to its effect on asymmetric histone inheritance within the GSC, we expressed histone H3T3A-GFP in differentiating cells using the *bamGal4* driver, which is expressed later in germline development and thus should not result in perturbation of asymmetric histone H3 inheritance in the GSC and GB[57]. We performed DNA FISH on *bam > H3-GFP* and *bam > H3T3A-GFP* and found no significant changes in the unpairing pattern in differentiating germ cells expressing histone H3T3A (Supplementary Fig. 5d–f), confirming that the perturbation in pairing states seen in *nos > H3T3A-GFP* is due to disruption of asymmetric histone H3 distribution in GSC and GBs.

To further confirm the effect of asymmetric histone inheritance on pairing of the *Stat92E* locus, we knocked down two genes reported to be required for this process. Haspin is a Serine/Threonine kinase that phosphorylates Thr3 on histone H3 and its

RNAi disrupts asymmetric histone inheritance[21]. Chromosome alignment defect 1 (Cal1) is required for asymmetric segregation of sister chromatids, each incorporated with old vs. new histone H3 and H4[58]. Therefore, knockdown of either gene should result in the same consequence of randomized histone inheritance. Consistent with this, we observed the *Stat92E* locus remaining paired throughout differentiation in temperature-sensitive *nosGal4*-driven *haspin* RNAi (Fig. 7h, i, k) and *Cal1 RNAi* testes (Fig. 7h, j, k). Cal1 has been shown to be required for centromere pairing in meiosis[59]. However, the *Stat92E* locus in GB and SG stages were more paired in *Cal1 RNAi* testes, indicating that the pairing defect in Cal1 knockdown is not due to a centromere pairing defect, but likely due to a histone inheritance defect.

These data suggest that the change in pairing state observed in GSC differentiation is epigenetically programmed during asymmetric stem cell division via stereotypic inheritance of histone H3.

**Stat92E is also regulated post-transcriptionally**. Finally, we asked whether the observed *Stat92E* pairing defect has any impact on Stat92E protein distribution. In wild type testes, Stat92E protein level shows a clear reduction in the GB and is almost non-detectable in the SG stage (Supplementary Fig. 6a[25,60]), even earlier than transcription shuts off (Supplementary Fig. 6b), indicating that the level of Stat92E is also regulated post-transcriptionally. We examined the pattern of Stat92E protein distribution in genotypes in which *Stat92E* pairing was perturbed. As expected, in all genotypes, Stat92E protein showed normal downregulation as seen in the wild type condition, with high expression in GSCs and immediate reduction in differentiating germ cells (Supplementary Fig. 6c–j). It has been shown that the defect in asymmetric histone inheritance disturbs germline

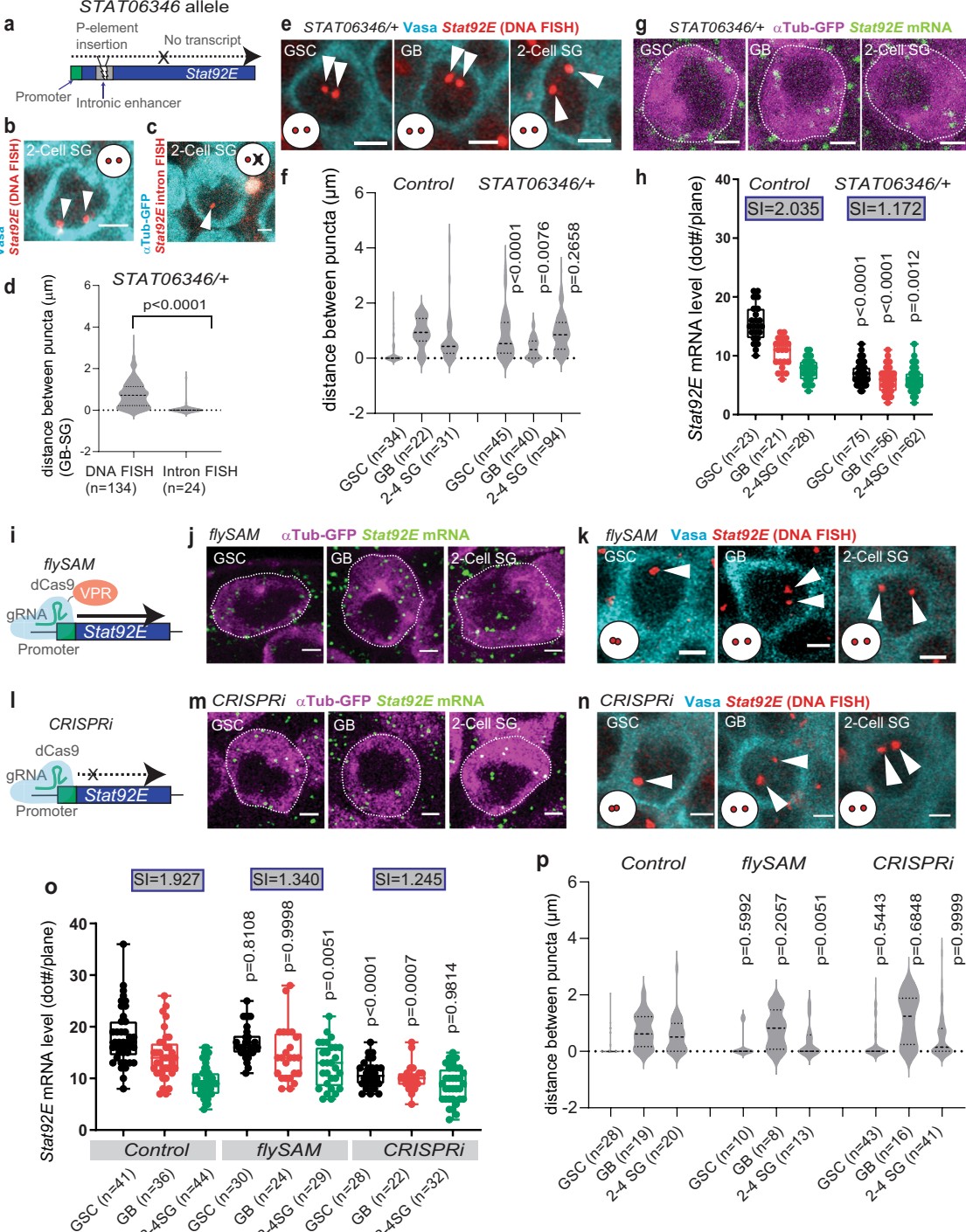

differentiation[21]. Therefore, we speculate that other genes that contribute to the differentiation of GSCs may be similarly regulated through pairing change.

Taken together, our data are consistent with a model whereby *Stat92E* downregulation occurs under the control of pairing regulation, providing a paradigm for how trans-chromosomal interactions mediate prompt gene downregulation during cell differentiation.

## Discussion

When homologs are close together, their proximity could regulate local transcription by trans-homolog regulatory mechanisms as

observed in transvection phenomena[9–14]. How the interaction between homologous regions influences local transcriptional activity and whether it occurs in endogenous gene regions have not been well understood. In this study, we demonstrate that the *Stat92E* gene is quickly downregulated during differentiation of the *Drosophila* male germline. The *Stat92E* allele is strongly paired in GSCs, and immediately becomes unpaired in GB following the asymmetric division (Fig. 6l). Disturbance of this pairing change results in a failure to quickly downregulate *Stat92E* expression, suggesting that the pairing change is required for switching transcriptional states. Given that enhanced (flySAM) and inhibited (CRISPRi) gene expression in cis did not affect pairing states, we propose that transcription is a

**Fig. 6 Regulation of pairing requires the *Stat92E* enhancer but not transcription. a** A schematic of the *Stat92E*[06346] allele. **b, c** Representative patterns of *Stat92E* DNA FISH (red, white arrowheads) (**b**) or nascent transcript (**c**) in 2-cell SGs (cyan) in *Stat92E*[06346]/+ flies. **d** Measured distances between puncta using DNA FISH or intron RNA FISH of *Stat92E*[06346]/+. **e** Representative images of *Stat92E* DNA FISH (red, white arrowheads) in *Stat92E*[06346]/+. Vasa (cyan). **f** Violin plots show distances between *Stat92E* DNA FISH puncta in *Stat92E*[06346]/+. **g** Representative images of *Stat92E* mRNA FISH (green) in *Stat92E*[06346]/+. *nos > αTubulin-GFP* (magenta). **h** Quantification of *Stat92E* mRNA levels in *Stat92E*[06346]/+. Y axis values are the number of mRNA dots present in a middle plane of the cell (dot#/plane). SIs (see text) are shown in boxes above bars. **i, l** Experimental design of the flySAM (**i**) and the CRISPRi (**l**). **j, m** Representative images of *Stat92E* mRNA (green) in *nos > flySAM* (**j**) or *nos > CRISPRi* (**m**). *nos > αTubulin-GFP* (magenta). **o** Quantification of *Stat92E* mRNA levels in control (*nos > αTub-GFP*), *nos > flySAM* or *nos > CRISPRi*. Y axis values are the number of mRNA dots present in a middle plane of the cell (dot#/plane). SIs (see text) are shown in boxes above bars. **k, n** Representative images of *Stat92E* DNA FISH (red, white arrowheads). **p** Violin plots of distances between DNA FISH puncta. All scale bars represent 2 μm. Number of scored cells, which are randomly chosen from at least 10 testes for each experiments, is shown for each data point. Violin plots show KDE and quantile lines and the width of each curve corresponds with the frequency of data points. Box plots show 25–75% (box), median (band inside) and minimum to maximum (whiskers) with all data points. All plotted data points are provided in Source Data. Representative pairing states are shown in lower left corner of each image. The *p* values were calculated by two-tailed Student's t-tests for **d**, and the adjusted *p* values were calculated using Šidák's multiple comparisons for other graphs. All scale bars represent 2 μm.

consequence, not cause, of local pairing regulation. Finally, we show that asymmetric histone inheritance dictates the *Stat92E* pairing change, indicating that the pairing change is epigenetically programmed during asymmetric division.

After asymmetric division, the GSC and GB still share almost identical intracellular and nuclear environments. The GB is displaced away from the niche and thus receives less niche signal (reviewed in[61]).This signal gradient that is initially present in the two daughter cells is quite shallow, and how the different fates of the two daughters are established remains to be determined. We propose the possibility that the physical separation of homologous regions initiates the change in gene activity, even when both cells are still exposed to similar signaling environments.

Disruption of either *Stat92E* pairing or unpairing did not change the level of *Stat92E* mRNA in GSCs, indicating that pairing condition is not simply activating transcription and unpairing is not simply repressing transcription (Figs. 4e, k and 6g). However, when the *Stat92E* expression level changes from a high to low level, it requires the change of pairing state, as evidenced by our data showing prompt reduction of expression is consistently disturbed in all genotypes with a pairing defect. The mechanism through which the *Stat92E* pairing change facilitates the downregulation of *Stat92E* expression in the *Drosophila* germline remains to be determined. A recent study demonstrated that paired homologous alleles can share common transcriptional resources, called a trans-homolog hub, which serves as a scaffold for the accumulation of transcription complexes during transvection[62]. If paired *Stat92E* alleles share a trans-homolog hub, the pairing-to-unpairing change may promote disassembly of the hub, which may facilitate prompt downregulation of expression. Further studies will be necessary to test this intriguing possibility.

Sister chromatids, with each containing either old or new histones H3 and H4, are inherited to the GSC or GB, respectively. These sister chromatids are hypothesized to have distinct epigenetic information for subsequent cell fate determination[20]. Perturbation of asymmetric histone H3 inheritance results in differentiation defects[21,63], which suggests that the preferential incorporation of new histones in the GB may be the mechanism actively erasing pre-existing epigenetic memory. Our data suggest that the observed change of *Stat92E* pairing is under the control of biased segregation of sister chromatids, suggesting an interesting possibility that the pairing may transduce inherent epigenetic information to actual gene expression states. The mechanism in which asymmetric inheritance of histones influences pairing states is unknown. Intriguingly, a recent study suggested that the GSC and GB enter S-phase with distinct timing[64]. As a result, two sister chromatids inherit different levels of chromatin condensation, which is required for distinct cell fate decision[64]. This suggests that distinct timing of nucleosome

assembly may determine the global epigenetic landscape differently in the GSC and the GB, and there is the possibility that the pairing states of *Stat92E* may be regulated by this process. Future studies determining the dynamic assembly of pairing/anti-pairing factors during S-phase may help to understand the mechanism. Moreover, it would be interesting to assess whether dedifferentiated GSCs, which are GSCs that have returned to the niche after differentiating into GBs or SGs, still retain the correct pattern of pairing/unpairing.

The mechanism in which interchromosomal interaction affects gene expression is not fully understood. In conventional gene regulation, local chromatin activity is regulated by active or repressive histone marks. Histone modifiers (writers and readers) reinforce each other through various feedback mechanisms to influence local gene activity (reviewed in[65]). Thus, there is a possibility that homologous gene regions can also influence each other's chromatin states when they are located in close proximity. It will be an interesting future study to define what types of chromatin modifications are shared by two homologous *Stat92E* alleles during the cell fate switch.

Homologous allelic pairing in a stem cell system was also reported for the *Oct4* locus in mice, where alleles of *Oct4* transiently pair in embryonic stem cells, likely to share repressive chromatin marks between homologous alleles during the transition from pluripotency to lineage commitment[66]. It is possible that pairing regulation may be commonly utilized for the change of key stem cell factors during differentiation of stem cells. Comprehensive, genome-wide analysis of interchromosomal interaction at other gene loci will be informative in this regard in the future.

In summary, our work provides evidence for the requirement of interchromosomal regulation for a switch in transcriptional state. We propose a model in which separation of homologous gene regions may be required for severing a trans-homolog effect to enable a rapid change in transcriptional activity even before the intracellular (or nuclear) environment changes. Such regulation could be a conserved mechanism for prompt downregulation of gene expression status during cell differentiation.

## Methods

**Fly husbandry and strains**. All fly stocks were raised on standard Bloomington medium at 25 °C (unless temperature control was required), and young flies (0- to 7-day-old adults) were used for all experiments. The following fly stocks were used: *nosGal4dVP16*[67]; *nosGal4VP16*[68]; *UAS-H3-GFP*[21]; and *UAS-H3T3A-GFP*[21]; *UAS-GFP-αTubulin* were gifts from Yukiko M. Yamashita; *tubGal80*[ts69], gift from Cheng-Yu Lee); *bamGal4VP16* (gift from Michael Buszczak). *lacO 98F6/99A77* (gift from Kristen Johansen); *hs-FLP; Ubi-GFP, FRT82B/FRT82B, P[PZ] Stat92E*[06346] (gifts from Erika Bach). For all RNAi experiments, *nosGal4, tub-Gal80*[ts] was used to induce short hairpin RNA expression for 5-7 days (or 3 days for *Cal1* RNAi to avoid germ cell loss) at 29 °C. To induce *P[PZ]Stat92E*[06346]

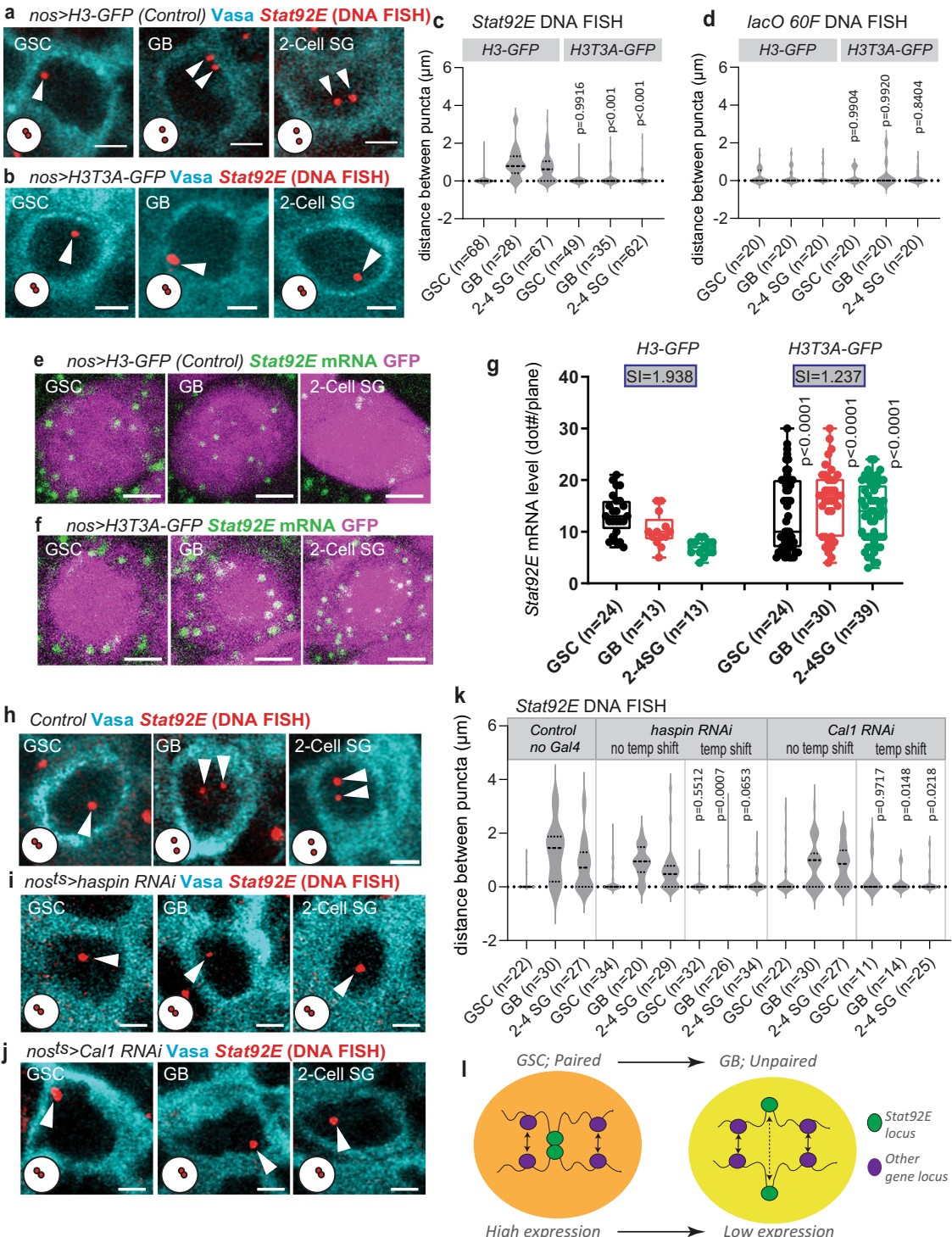

mutant clones, flies were heat shocked at 37 °C once for 1 hour, then dissected after 1 day.

Other stocks were from Bloomington Stock Center (BDSC): *lacO 60 F* (BDSC 25371), *Df(3 R)BSC516* (BDSC 25020), *P[PZ]Stat92E*[06346] (BDSC 11681), *Stat92E* BacTg (*Stat92E-GFP.FLAG, VK00037*) (BDSC 38670), *SAM.dCas9.GS02442* (BDSC 80517), *Stat92E*[TOE.GS02090] (BDSC 80745), *Mrg15 RNAi TRiP.GL00128* (BDSC 35241), *Slmb RNAi TRiP.JF01504* (BDSC 31056), *haspin RNAi TRiP.GL00176* (BDSC 35276), *Cal1 RNAi TRiP.HMS02281* (BDSC 41716), and Fly-Fucci; *UASp-GFP.E2f1.1-230* (BDSC 55100)[49].

**Generation of UASP-dCas9-mCherry transgenic fly**. pWalium20-10XUAS-3XFLAG-dCas9-VPR vector (Addgene) was digested by NheI and SphI. dCas9 was amplified from same vector using following primers; NheI Cas9m4 F; 5′-CCATA AAACATCCCATATTCAGC-3′ Cas9m4-NLSR; 5′-AGCCCGTCCGGAACC

GCTGGCCTC-3′. mCherry was amplified from pmCherry-C1 vector using following primers; mCherryF; 5′-GACGCCAGCGGTTCCGGACGGGCTGTGAG-CAAGGGCGAGGAGGATAACA-3′; SphI mCherryR; 5′-GGACAGTCCTG TGCTGATATGCATGGCATGCCTTGTACAGCTCGTCCATGCCGCCGGT-3'. Obtained PCR products with digested vector were assembled by Gibson assembly (NEB) following manufacturer's instruction. Resultant plasmid was verified by sequencing and transgenic flies were generated using strain attP2 by PhiC31 integrase-mediated transgenesis (BestGene Inc.).

**Immunofluorescence staining**. Testes were dissected into 1X phosphate-buffered saline (PBS) and fixed in 1 ml of 4% formaldehyde in PBS for 30-60 minutes, then washed three times in 1 ml of PBS + 0.3% TritonX-100 (PBST) for one hour, then incubated in primary antibodies in 100 µl of 3% bovine serum albumin (BSA) in PBST at 4 °C overnight. Samples were then washed three times in 1 ml of PBST for

**Fig. 7 Stat92E pairing is under the control of asymmetric histone inheritance. a, b** Representative images of *Stat92E* DNA FISH (red, white arrowheads) at the indicated stages of germ cells from *nos > H3-GFP* (**a**) or *nos > H3T3A-GFP* (**b**). Vasa (cyan). **c, d** Violin plots of distances between *Stat92E* DNA FISH puncta (**c**) or between the *lacO 60 F* locus in *lacO 60 F* homozygous flies, both expressing *nos > H3* or *nos > H3T3A*. **e, f)** Representative images of *Stat92E* mRNA (exon probe smFISH; green) at the indicated stages of germ cells from *nos > H3-GFP* (**e**) or *nos > H3T3A-GFP* (**f**). GFP (magenta). **g** Quantification of *Stat92E* mRNA levels in control (*nos > H3-GFP*) or *nos > H3T3A-GFP* expressing germ cells. Y axis values are the number of mRNA dots present in a middle plane of the cell (dot#/plane). SIs (see text) are shown in boxes above bars. **h, i, j** Representative images of *Stat92E* DNA FISH (red, pointed by white arrowheads) at the indicated stages of germ cells from control (**h**), *haspin* RNAi (**i**) or *Cal1* RNAi (**j**) testes. Vasa (cyan). **k** Violin plots of distances between DNA FISH puncta. Temperature-sensitive *nosGal4*[ts] driver was used for all RNAi experiments. Temperature shift was performed in 29 °C for three days for *Cal1* RNAi to avoid germ cell loss phenotype, five days for *haspin* RNAi. *P* values were provided by comparison with no-temp-shift controls. **l** A schematic shows observed change of local pairing states of the *Stat92E* gene and effect on gene silencing. Violin plots show KDE and quantile lines and the width of each curve corresponds with the frequency of data points. Box plots show 25–75% (box), median (band inside) and minimum to maximum (whiskers) with all data points. All plotted data points are provided in Source Data. Number of scored cells, which are randomly chosen from at least 10 testes for each experiments, is shown for each data point. For all graphs, adjusted *p* values were calculated using Šidák's multiple comparisons. All scale bars represent 2 µm. Representative pairing states are shown in lower left corner of each image.

one hour (three 20 min washes), then incubated in secondary antibodies in 100 µl of 3% BSA in PBST for 2–4 h at room temperature, or at 4 °C overnight. Samples were then washed three times in 1 ml of PBST for one hour (three 20 minute washes), then mounted in a drop of VECTASHIELD with 4,6-diamidino-2-phenylindole (DAPI) (Vector Lab, H-1200).

Primary antibodies used were: guinea pig anti-Stat92E[67] (1:2000), gift from Yukiko Yamashita), rat anti-Vasa (DSHB, anti-vasa/AB_760351, developed by A. Spradling and D. Williams, 1:20), rabbit anti-Vasa (d-260, Santa Cruz Biotechnology, Santa Cruz, CA, 1:200) and guinea pig anti-Traffic jam (Tj)[70] (1:5000) (gift from Dorothea Godt). AlexaFluor-conjugated secondary antibodies were used at a dilution of 1:400. Secondary antibodies used were Goat Anti-Rabbit IgG H&L (Alexa Fluor 488, Abcam, ab175652), Goat Anti-Rabbit IgG H&L (Alexa Fluor 647, Abcam, ab150079), Goat Rat IgG H&L (Alexa Fluor 488, Abcam, ab150157), Goat Rat IgG H&L (Alexa Fluor 647, Abcam, ab150159), and Goat Anti-Guinea Pig IgG H&L (Alexa Fluor 647, Abcam, ab150187).

**RNA fluorescence in situ hybridization**. Fluorescence in situ hybridization was performed as described previously. Briefly, testes were dissected in 1X PBS and then fixed in 1 ml of 4% formaldehyde/PBS for 45 minutes. Fixed testes were rinsed 2 times with 1 ml of 1X PBS, then resuspended in 1 ml of ice-cold 70% EtOH, and incubated for 1 hour-overnight at 4 °C. Testes were rinsed briefly in 1 ml of wash buffer (2X SSC and 10% deionized formamide), then incubated overnight at 37 °C for 16 hours in the dark with 50 nM of Stellaris probes in 200 µl of Hybridization Buffer containing 2X SSC, 10% dextran sulfate (MilliporeSigma), 1 µg/µl of yeast tRNA (MilliporeSigma), 2 mM vanadyl ribonucleoside complex (NEB), 0.02% RNase-free BSA (ThermoFisher), and 10% deionized formamide. Then, testes were washed 2 times for 30 minutes each at 37 °C in the dark in 1 ml of prewarmed wash buffer (2X SSC, 10% formamide) and resuspended in a drop of VECTASHIELD with DAPI. Quasar 570 labeled Stellaris probe against a third intron sequence of *Stat92E* gene and Quasar 670 labeled Stellaris probe against a second exon of *Stat92E* gene were obtained from LGC Bioseach Technologies (target sequences are provided in Supplementary Data 1). Quasar 570 labeled Stellaris probe against the *nanos* 3'UTR sequence was gift from Michael Buszczak.

**Detection and quantification of mRNA by smFISH**. For visualization of germ cells, *nos > αTubulin-GFP*, *nos > histone H3-GFP* were used. For experiments using *BacTg* (*Stat92E-GFP.FLAG*, *VK00037*), Stat92E-GFP signal was used to visualize germline and CySCs. Single molecule FISH was performed using the method described above, with a Quasar 670 labeled Stellaris probe set targeting a second exon of *Stat92E* (Supplementary Data 1) or 3'UTR region of *nanos* mRNA. Imaging was performed by using a Zeiss LSM800 airy scan with a 63× oil immersion objective (NA = 1.4) and processed using ImageJ/Fiji.

Because germ cells and somatic CySCs or CCs, which are both positive for *Stat92E* mRNA, closely adhere to each other, *Stat92E* smFISH signal in germ cells and in somatic cells overlap at the surface of cells (main figure, Fig. 1g inset). To avoid mistakenly counting the smFISH dots localized in somatic cells, we analyzed approximately mid-plane of each germ cell. To confirm this method can accurately reflect mRNA levels, we performed a smFISH using a germline specific gene, *nanos*, which expressed specifically in germline (Supplementary Fig. 1c). Using the *nanos* smFISH, we compared scoring results obtained from whole cell (Supplementary Fig. 1d) vs. single plane (Supplementary Fig. 1e). *Nanos* smFISH showed a similar downregulation pattern to that of *Stat92E* and two methods (single plane scoring vs. whole cell scoring) showed almost identical silencing indices (SIs, ratio of smFISH number of stages, GSC/2-4 SG) in c and d. Therefore, we concluded that SI could be reliably used for comparison of *Stat92E* downregulation rates across different genotypes.

smFISH signal appeared as uniform size and intensity of dots in entire cytoplasm. We observed brighter dots that likely are the detection of multiple overlapping mRNA molecules. Because such cases were rare (~1% for *Stat92E*, ~5%

for *nanos*), we considered the frequency to have negligible effect on the total number and therefore these dots were also counted as 1.

Our quantification of *Stat92E* mRNA levels showed a linear correlation with *Stat92E* copy number (Supplementary Fig. 1h), suggesting that the method used for scoring was quantitative. Note that the obtained SIs were different in each genotype, likely reflecting the effect of interchromosomal interaction among the genotypes containing different copy numbers of *Stat92E* alleles.

**Detection and quantification of nascent transcript**. Nascent transcript was visualized by RNA FISH method described above, with a Quasar 570 labeled Stellaris probe set targeting a third intron of *Stat92E* gene together with a Quasar 670 labeled Stellaris probe against a second exon of *Stat92E* gene. 0.5 µm or 1 µm interval z-stack was taken for entire testis tip area using a Zeiss LSM800 airy scan with a 63× oil immersion objective (NA = 1.4). Images were processed using ImageJ/Fiji. *Stat92E* nascent transcript was seen as one or two foci in each nucleus double-positive with both of intron and exon probes. For intensity measurement, one to three z-stacks for the entire area of a punctate were integrated after background levels were subtracted. To compare signal between samples, CySCs were used for internal control, which typically show uniform intensity of *Stat92E* nascent transcript within a testis tip. Measured intensities of germ cells were divided by average intensity of 2 or 3 randomly picked paired CySCs. Background level was subtracted for each measurement. Paired cases were judged when intron FISH signal was present as a single spot and no other spots were found in the same nucleus. Measured intensities of paired foci were divided by two and plotted as intensity/allele.

**OligoPaint probe production**. OligoPaint probes were designed using PaintSHOP online software[71] with the dm6 genome. The *Stat92E* sense probe set consisted of 949 oligos targeting the *Stat92E* locus and surrounding regions, from 3 R:20,510,024..20,569,794 (Supplementary Data 2). Each oligo consisted of a region complementary to a genomic region of the sense strand of the *Stat92E* locus, flanked by a secondary recognition site (Sec5) on the 5' end, and a T7 site on the 3' end (for example: Sec5:AGCGCAGGAGGTCCACGACGTGCAAGGGTGTttt… Genomic target:ACCTGCTCCAGGTGCTTGCCGTTCTTCGGATTTatcg… T7 site:tctcccTATAGTGAGTCGTATTA), (Twist Bioscience). Oligos were amplified twice by PCR (Phusion High-Fidelity DNA Polymerase, NEB) following manufactural instruction, using the following primers:

Forward: 5′-AGC GCA GGA GGT CCA CGA CGT GCA AGG GTG-3′
Reverse: 5′-TAA TAC GAC TCA CTA TAG GGA GAC GAT-3′
(Integrated DNA Technologies IDT).

PCR product was purified using Oligo Clean & Concentrator Kits (Zymo Research, D4060). RNA was synthesized from 700 ng of amplified oligos using T7 RNA polymerase (HiScribe T7 kit, NEB) following manufacturer instruction. RNA product was then reverse transcribed to cDNA using Maxima H Minus Reverse Transcriptase (Thermo Fisher Scientific). Briefly, 15 µl of 100 µM forward primer and 24 µl of 10 mM each dNTPs (Thermo Fisher Scientific), 30 µl of 5X buffer and 57.5 µl of water were added to 20 µl of RNA product then incubated in 65 °C for 5 min for denaturing. 1.5 µl of RNaseOUT (Thermo Fisher Scientific) and 2 µl of Reverse Transcriptase were added and incubated in 50 °C for 2 hours. Template RNA was removed by alkaline hydrolysis adding 150 µl of 1:1 mixture of 0.5 mM EDTA and 1 M NaOH, incubated in 95 °C for 10 min. Resultant single-stranded oligos were purified by using Zymo DNA concentration kit (Zymo Research, D4030) modified for short-length DNA cleaning. Briefly, 600 µl of Oligo binding buffer (Zymo Research, D4060-1-40) and 2400 µl of ethanol were added to sample then loaded onto column and the centrifuge method was followed as per the manufacturer's instruction. Purified probe was quantified with nanodrop (Thermo Fisher Scientific) and 200 pmol of probes were used for each hybridization reaction.

*Stat92E* antisense probes were produced using the amplified PCR product of the *Stat92E* sense pool as a template. Antisense oligos were amplified by PCR using

oligos to add a secondary site (Sec4) to the 5′ end and a Sp6 site on the 3' end. The following primers were used for amplification:

Forward: ACCCGCAGGACACCTAACCCGTCACCGTCCGATTTTTTTTTTGGAATTG TGAGCGGATAACAATT

Reverse:CCCGCAGGACACCTAACCCGTCACCGTCCGACGACTCACTAT AGGGAGACGAT

The same process was followed as described for the sense pool production, using HiScribe Sp6 kit (NEB) instead of T7.

The secondary probes were designed to be complementary to the secondary sites, with fluorophores on both 5' and 3' ends of the oligo (IDT).

Sec4 Secondary: Cy3- TCGGACGGTGACGGGTTAGGTGCCTGCGGG -Cy3
Sec5 Secondary: Cy5- AACACCCTTGCACGTCGTGGACCTCCTGCGCTA -Cy5
Sec5 sequence: AGCGCAGGAGGTCCACGACGTGCAAGGGTGT
Sec4 sequence: CCCGCAGGACACCTAACCCGTCACCGTCCGA
lacO probe sequence: TGGAATTGTGAGCGGATAACAATT

**DNA fluorescence in situ hybridization and immunofluorescence**. Testes were dissected into PBS and processed for immunofluorescence with rabbit anti-Vasa and mouse anti-Hts antibodies as described above. After incubation in secondary antibodies, testes were washed three times (20 min each) in 1 ml of 0.3% PBST and then post-fixed for 10 min in 1 ml of 4% formaldehyde/PBS. Testes were then rinsed in 1 ml of 2X SSC (20XSSC was obtained from Thermo-Fisher) with 0.1% Tween-20 (Thermo-Fisher) (SSCT), three times for 3 min each. To allow a gradual transition into 50% formamide, testes were washed for ten minutes each in 1 ml of 20%, 40%, then 50% formamide in 2X SSCT. Testes were then heat denatured at 92 °C for 30 min in 1 ml of 50% formamide in 2X SSCT, and incubated in the probe mix at 37 °C for 16 h. The probe mix consisted of 50% formamide and 10% dextran sulfate (Sigma-Aldrich) in 2X SSCT with 200 pmol of primary oligos, 100 pmol of secondary oligos, and 2.5 µl of RNase Cocktail Enzyme Mix (Thermo-Fisher). The probe mix (200 µl/sample) was denatured at 65 °C for 5 min and kept on ice before adding to the samples. After the incubation, 1 ml of 50% formamide/2X SSCT was added to the sample then removed with the probe mix. Samples were washed again with 1 ml of 50% formamide/2X SSCT for 1 h, then in 1 ml of 20% formamide/2X SSCT for 10 min, all at 37 °C. Finally, samples were washed two times, with 1 ml of 2X SSCT for 10 min each at room temperature and mounted in a drop of VEC-TASHIELD with DAPI. Imaging was performed on a Zeiss LSM800 confocal microscope with a 63× oil immersion objective (NA = 1.4) and processed using airy scan and ZEN software.

**Measurement of distance between homologous loci**. To measure distances between DNA FISH or RNA intron FISH puncta, we imaged a z-stack of the entire testis tip with optimized interval (0.5 µm to 1 µm) by using a Zeiss LSM800 airy scan with a 63× oil immersion objective (NA = 1.4). Using ImageJ/FIJI software, we judged whether a cell has single punctate (paired) or two separate puncta (unpaired) pattern. For all paired cases in which punctate appears as a single spot, we plotted zero on each violin plot. When two signals were located in different z-stacks, we calculated the distance using an equation,

$$\mathbf{D} = \sqrt[2]{x^2 + z^2} \quad (1)$$

where $\mathbf{D}$ is 3D distance between punctae, $x$ is a measured 2D distance on a single plane and $z$ is z-stack distance, by Microsoft Excel and plotted values on each violin plot. Paired punctae had approximately twice as much intensity as unpaired punctae (see Fig. 4p) and when the intensity was visibly off from the range, the punctate was omitted from measurement. Cells with more than three puncta were also omitted from measurement. 3D distance scoring was confirmed by using Imaris 9.5 (Oxford Instruments Group, see more details in Supplementary Fig. 2).

**Detection of S-phase germ cells**. S-phase detection was performed using Click-iT™ EdU Cell Proliferation Kit for Imaging, Alexa Fluor™ 594 dye (Thermo-Fisher, C10337) following the manufacturer's instructions. Briefly, testes were dissected in PBS then transferred to Schneider's media. 10 µM 5-ethynyl-2'-deoxyuridine (EdU) was added to the media and incubated for 2 hours at room temperature to allow EdU to incorporate. Testes were then washed three times for five minutes each in 1% BSA in PBS and fixed in 4% formaldehyde in PBS for 15 min at room temperature, and washed three times for 5 min each in 1% BSA in PBS. Testes were then resuspended in Click-iT™ Wash and Permeabilization Buffer for 10 min and then incubated with reaction cocktail for 30 min in room temperature. Testes were then washed three times for five minutes each in Click-iT™ Wash and Permeabilization Buffer, and fixed for 30 min in 4% formaldehyde in PBS then washed three times in PBS + 0.3% TritonX-100 (PBST) for 1 h. Samples were then incubated in primary antibodies in 3% bovine serum albumin (BSA) in PBST at 4 °C overnight and washed three times in PBST for 1 hour (three 20 min washes), then incubated in secondary antibodies in 3% BSA in PBST for 2–4 h at room temperature. Samples were then washed three times in PBST for 1 h (three 20 min washes), then mounted using VECTASHIELD with 4,6-diamidino-2-phenylindole (DAPI) (Vector Lab, H-1200). Antibodies used for immunofluorescence were rabbit anti-Vasa (d-260, Santa Cruz Biotechnology, Santa Cruz, CA, 1:200) and Goat Anti-Rabbit IgG H&L (Alexa Fluor 647, Abcam, ab150079, 1:400).

**TAD boundary identification**. We examined the published Hi-C sequencing data with topologically associating domain (TAD) coordinates in *Drosophila* larval eye discs at 10 kb resolution[7] (GEO: GSE136267). TAD calls were identified by the original paper, and the details are shown below: TAD calls were based on a Hidden Markov Model (HMM) segmentation of the DI scores. The HMM was initialized with three states (downstream bias, neutral, upstream bias), each with a three-part equally Gaussian mixture model. TADs were defined as starting at the first downstream bias state following an upstream bias state with any number of intervening neutral states. We confirmed that the *Stat92E* gene is located within the TAD at chr3R:20470000~20570000.

**Statistics and reproducibility**. No statistical methods were used to predetermine sample size. The experiments were not randomized. The investigators were not blinded to allocation during experiments and outcome assessment. For all violin plots, distances between two puncta of OligoPaint DNA FISH signal were plotted. Cells with more than three dots were omitted from measurement (see details in Supplementary Fig. 2). Violin plots show KDE and quantile lines and the width of each curve corresponds with the frequency of data points. All box plots show 25–75% (box), median (band inside) and minimum to maximum (whiskers) with all data points. The $p$ values (two-tailed Student's $t$-test or adjusted $p$ values from one-way anova test) are provided (comparison between each genotype with wild type data shown in Fig. 2c or control data shown in the same graph unless otherwise indicated. Statistical analysis and graphing were performed using GraphPad Prism 9 or Microsoft excel software. All plotted data points are provided in Source Data. Experiments were repeated 2 times for Fig. 3c and Supplementary Fig. 5c, or three times for other figures to confirm results. Number of scored cells, which are randomly chosen from at least 10 testes for each experiments, is shown in each graph.

**Reporting summary**. Further information on research design is available in the Nature Research Reporting Summary linked to this article.

## Data availability
The data that support this study are available from the corresponding author upon reasonable request. The confocal image data generated in this study have been deposited in the BioStudies database under accession number S-BSST829. Data used to analyze TAD boundaries were accessed by the GEO database under accession number GSE136267. Source data are provided with this paper.

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

## Acknowledgements

We thank Jelena Erceg, Yukiko M. Yamashita for valuable discussions; Marie Bao for manuscript editing; Erika Bach, Yukiko M. Yamashita, Michael Buszczak, Cheng-Yu Lee, Kristen Johansen, the Bloomington *Drosophila* Stock Center and the Developmental Studies Hybridoma Bank for reagents. This research is supported by R35GM128678 (to M.I.), R35GM133562 (to S.L.) from the National Institute for General Medical Sciences and start-up funds from UConn Health (to M.I.).

## Author contributions

M.I. and M.A, conceived the project, designed and executed experiments and analyzed data. M.M. and R.R. executed experiments and analyzed data. B.M. provided the methods/materials and assisted with the design of the OligoPaint DNA FISH experiments. Z.P. and S.L. analyzed published Hi-C data. All authors wrote and edited the manuscript.

## Competing interests

The authors declare no competing interests.
