## [Peer Review File · Nature Communications]

REVIEWER COMMENTS

Reviewer #1 (Remarks to the Author):

This is an interesting paper that describes an unusual type of gene regulation during cell differentiation through homologous chromosome pairing, a mechanism which the authors propose to be common for differentiation-specific gene regulation specifically at the transcriptional level. The interaction and subsequent loss of pairing between homologous alleles of Stat92E upon GSC differentiation is presented as the reason for the gene's repression in the differentiating GB. The authors show the asymmetric inheritance of histones H3 and H4 as the driving factor for the loss of interchromosomal pairing in differentiated GBs, requiring the Stat92E enhancer element but independent of the gene's transcription.

Major:

1. Figure 1D: How are the distances measured. Because the cells are in 3D, how did the author ensure that they are measuring the distance between 2 puncta accurately if they are looking at a single plane or projection? In addition, Fig. 1D is not called out in the text until after Fig. 2A. It will be good to arrange the figures in the same sequence that they are being called out in the text.

2. What is the biological significance of such a regulation? Figure 1E,F: The authors show that Stat92E mRNA gradually decreases during differentiation. In Fig. S1A, they show that STAT92E protein is immediately downregulated in gonialblast. The authors also suggest that the levels of Stat92E mRNA and protein do not correlate well and the regulation of Stat92E transcription may not play a crucial role? Instead, translation and/or protein degradation may be more important in the regulation of Stat92E during GSC-GB differentiation as the authors also suggested. For those instances where the authors show stabilization of Stat92E mRNA during differentiation, it would be good to also show the Stat92E protein levels and the differentiation defects (if any).

3. Measurement of intron RNA intensity. How quantitative is the signal measurement? Can the authors provide a control measuring the intron RNA intensity in different genetic background with 0, 1, 2, 3 or 4 copies of stat92E and plot a standard curve? This point is related to Fig. 3M, N, where the authors suggest that paired loci have increased transcription. Another interpretation is that colocalization of 2 puncta leads to a $1+1=2$ signal intensity consistent with their data. If there is an increase in transcription, I would expect an even higher intensity level than 2-fold.

Minor:

1. P-values are missing in Fig. 4O,P
2. High background in Fig. 3I, 5B,H, S2C
3. Spelling in Fig. 4A

Reviewer #2 (Remarks to the Author):

In this work, Antel and Inaba use the *Drosophila* male germline stem cell system to study the molecular mechanisms between homologous chromosome pairing, in particular at the Stat92E gene locus, and the transcriptional regulation, mainly the down-regulation from stem cell to more differentiated cells. The results demonstrate that the chromosomal/chromatin changes occur prior to the actual transcript change and likely contributes to the latter change during germline differentiation. These results should lead

to new information regarding the developmentally regulated key gene expression during stem cell differentiation and shed light on the downstream effects of asymmetric histone inheritance. This study also provides an example how intrinsic factors act to give rise to an immediate key status change, despite the shallow change of the extrinsic signaling by itself. In this work, the authors also used cutting-edge molecular genetics and cell biology techniques to visualize detailed changes from GSC to its immediate daughter GB cell and so on, therefore provide new tools to study these cells in this model system. Overall, the findings should be of great interest to the readers of Nature Communications. However, there are some concerns and suggestions about the logic flow and data presentation (see below for details when talking about individual parts of the Results). Hope these suggestions will help the authors to make the data presentation more convincing and compelling.

1. Figure 1: Stat92E exon smFISH gave out many "dots" everywhere, including nucleus, cytoplasm and extracellular regions. In particular, abundant signals were detected in the hub region. What is the control to make sure these signals are real? And how to reconcile these results, that Stat92E transcript is present in hub cells and becomes gradually lost during germline differentiation, with the previous results using the Stat reporters, for which I do not think such a broad expression pattern was observed? A technical question: For the quantification of Stat92E mRNA levels in 1F, would using dots/ \sim middle plane introduce any potential bias as it is not necessary that these signals are evenly distributed? Shouldn't the entire dots per cell be more accurate? Some dots are brighter and could they be overlapping multiple dots? The signal intensity could be more accurate if it is not saturated. Another technical question: Given that GSCs have elongated G2 phase, the signals in GSCs should reflect signals from duplicated chromosomes, but how about CySCs and other staged germ cells? In another word, how to avoid effect by different cell cycle stages among different cell types (i.e. germline vs. cyst) and different stages (i.e. stem cell vs. more differentiated cells)? In the figure legend of 1D, it is mentioned that "Cells with more than three puncta were omitted from scoring (see description about Fig 1H)", but I cannot find such a description for Fig. 1H. Do the authors mean Fig. S1B? Finally, for this figure, as the comparison among different cell types and stages needs internal control for technical reasons, such as FISH efficiency, it should be noted these Ns in 1D, 1F and 1H come from how many individual testes and from how many independent experiments.

2. Figure 2: The same technical question applies here: For direct comparison of FISH signals, how to avoid effect by different cell cycle stages among different cell types (i.e. germline vs. cyst) and different stages (i.e. stem cell vs. more differentiated cells)? Also, it should be noted the Ns in 2C, F, I, L, O, particularly in 2C, come from how many individual testes and from how many independent experiments. Finally, in 2M-O, in this deficiency line, how is the Stat92E mRNA level, does it stay high and if so, does this line have any phenotype by itself?

3. Figure 3-5: The same technical question: For the quantification of Stat92E mRNA levels in 3B, H, L, and 4H, L, as well as 5G, would using dots/ \sim middle plane introduce any potential bias? Shouldn't the entire dots per cell be more accurate? Some dots are brighter and could they be overlapping multiple dots? The signal intensity could be more accurate if it is not saturated. Also, if the "paired" situation correlates with higher intron smFISH signals compared to "unpaired", why does the "unpaired" situation in Bac Tg/Df show higher levels of signals in 3H and why do the nos>slmb RNAi ("unpaired") show higher levels of signals than those in the nos>mrg15 RNAi ("paired")? However, the "paired" situation in nos>H3T3A (paired) show higher levels of signals than that in nos>H3 (unpaired), consistent with the paired status with higher levels. Or, does it only matter for the change between "unpaired" and "paired"? And if it does, the mRNA levels do not seem to significantly change between wild-type GSC (mostly paired) and GB (mostly unpaired), as shown in Fig. 1F and 1H? I am trying to get the logic straightened here for these results, but it seems something is missing in this chain. Also, in Fig. 4N, does the GSC show a "paired" signal? If so, why does the cartoon show an "unpaired" scenario?

4. I wonder whether the histone status affect the Stat92E enhancer region, can this be tested using the nos>H3T3A with the STAT06346 allele? This could also be investigated using the reporters with the putative Stat-binding sites, given that Stat92E is self-regulated through a positive feedback. This would be very informative if this regulation is through Stat-binding sites, therefore paired situation could enhance the co-activation of both alleles through dimerization.

5. RNAi conditions for Fig. 3 and 5 need to be described in the Methods, for example, is it a constitutively knockdown or temperature-controlled conditional knockdown? This could be an issue to consider if prolonged knockdown leads to cellular defects.

6. Another technical question about measuring the distance between intron probe signals or between DNA FISH signals, how to measure their distances on the Z-axis?

7. This question is a bit remote which is not necessary to be addressed in this work but some discussion would be helpful: What do the authors think about the de-differentiated GSCs? Will they regain the pairing status or not? Another point that the authors could discuss more is whether these changes affect Stat92E's function as a transcription factor (i.e. protein level). For example, does failure in downregulate Stat92E transcript properly lead to tendency of perdurance of Stat92E protein? Finally, a preprint reported less condensed chromosomes in GB compared to GSC

(<https://doi.org/10.1101/2021.03.08.434490>), could this difference contribute to the change of pairing of homologous chromosomes from GSC to GB immediately after stem cell asymmetric division? Again, this could be beyond this paper but could be discussed and for future studies.

8. Minor point: Is reference 30 being mis-referred as it is a paper in Drosophila for the developmental roles of JAK-STAT but not on mammalian cells?

9. Minor point: In Discussion-- "The Stat92E pairing pattern was unlikely directly affected by the chromosome segregation defect caused by H3T3A expression as the mechanism of homolog pairing and sister chromatid cohesion seem to be distinct in Drosophila cells [59]." This sentence is a bit confusing, the H3T3A affects biased histone inheritance (i.e. sister chromatids carrying old vs. new H3/H4 are mis-inherited) but not chromosome segregation defect which often leads to aneuploidy, etc. Better to clarify this.

Point-by-point responses are provided below.
Reviewer comments are shown in **blue**, our responses are shown in **black**.

Reviewer #1 (Remarks to the Author):

This is an interesting paper that describes an unusual type of gene regulation during cell differentiation through homologous chromosome pairing, a mechanism which the authors propose to be common for differentiation-specific gene regulation specifically at the transcriptional level. The interaction and subsequent loss of pairing between homologous alleles of Stat92E upon GSC differentiation is presented as the reason for the gene's repression in the differentiating GB. The authors show the asymmetric inheritance of histones H3 and H4 as the driving factor for the loss of interchromosomal pairing in differentiated GBs, requiring the Stat92E enhancer element but independent of the gene's transcription.

First, we would like to thank this reviewer for his/her encouraging and insightful comments and great suggestions.

Major:

1. Figure 1D: How are the distances measured. Because the cells are in 3D, how did the author ensure that they are measuring the distance between 2 puncta accurately if they are looking at a single plane or projection? In addition, Fig. 1D is not called out in the text until after Fig. 2A. It will be good to arrange the figures in the same sequence that they are being called out in the text.

Thank you for raising this point. The other reviewer made a similar comment as well, and we realized that the method in which we measured between two puncta was not explained in the previous version of the manuscript. We added explanation and new **Supplementary Figure 2** to validate the method, and also detail the information below:

First, we took series of z-stacks from the testis tip with optimized interval (0.5 μ m to 1 μ m step). Then, we selected germ cells in each developmental stage based on their relative location to the hub and connection of cells. We judged whether each cell has a "paired" or "unpaired" pattern. For "paired" cases in which punctae appear as a single spot, we plotted "zero" on each violin plot. When two signals were located in different z-stacks, we calculated the distance by simple math: $(\text{actual distance})^2 = (\text{distance on a plane})^2 + z^2$, using excel and plotted the value on each violin plot. We updated Method section as follows:

"To measure distances between DNA FISH or RNA intron FISH puncta, we imaged z-stack of entire testis tip with optimized interval (0.5 μ m to 1 μ m step) by using a Zeiss LSM800 airy scan with a 63 \times oil immersion objective (NA=1.4). Using ImageJ, we judged whether a cell has single punctate "paired" or two separate puncta "unpaired" pattern. For all "paired" cases in which

punctate appears as a single spot, we plotted “zero” on each violin plot. When two signals were located in different z-stacks, we calculated the distance using an equation,

$$\text{Distance} = \sqrt{x^2 + z^2}$$

where D is distance between punctae, x is a measured 2D distance on a single plane and z is z-distance, by Microsoft Excel and plotted values on each violin plot.”

To validate that the method we used was accurately reflecting the distance between 2 alleles, we further performed a 3D measurement using Imaris software which offers tools for measurement simply by manually clicking two punctae in different plane of 3D-stack. We confirmed that there is no significant difference between our original measurements and Imaris measurements (new **Supplementary Figure 2**).

2. What is the biological significance of such a regulation? Figure 1E,F: The authors show that Stat92E mRNA gradually decreases during differentiation. In Fig. S1A, they show that STAT92E protein is immediately downregulated in gonialblast. The authors also suggest that the levels of Stat92E mRNA and protein do not correlate well and the regulation of Stat92E transcription may not play a crucial role? Instead, translation and/or protein degradation may be more important in the regulation of Stat92E during GSC-GB differentiation as the authors also suggested. For those instances where the authors show stabilization of Stat92E mRNA during differentiation, it would be good to also show the Stat92E protein levels and the differentiation defects (if any).

Thank you for bringing this up. We have examined Stat92E protein levels in the genotypes in which pairing was perturbed, and found no differences in protein expression pattern compared with the controls (**Supplementary Figure 6**). This is consistent with our data showing that Stat92E protein is also regulated post-transcriptionally (Original **Fig. S1A** and now in **new Supplementary Figure 6A, B**). Although, we selected *Stat92E* as a model to study pairing effect on transcription, we speculate that other genes, which may be similarly regulated through pairing change, may contribute to the differentiation of GSCs. Revised result section reads as follows:

“Finally, we asked whether the observed *Stat92E* pairing defect has any impact on Stat92E protein distribution. In wild type testes, Stat92E protein level shows a clear reduction in the GB and is almost non-detectable in the SG stage (**Supplementary Figure 6A** [25, 60]), even earlier than transcription shuts off (**Supplementary Figure 6B**), indicating that the level of Stat92E is also regulated post-transcriptionally. We examined the pattern of Stat92E protein distribution in genotypes in which *Stat92E* pairing was perturbed. As expected, in all genotypes, Stat92E protein showed normal downregulation as seen in the wild type condition, with high expression in GSCs and immediate reduction in differentiating germ cells (**Supplementary Figure 6C-J**). It has been shown that the defect in asymmetric histone inheritance disturbs germline differentiation [21]. Therefore, we speculate that other genes that contribute to the differentiation of GSCs may be similarly regulated through pairing change.”

3. Measurement of intron RNA intensity. How quantitative is the signal measurement? Can the authors provide a control measuring the intron RNA intensity in different genetic background with 0, 1, 2, 3 or 4 copies of *stat92E* and plot a standard curve?

This point is related to Fig. 3M, N, where the authors suggest that paired loci have increased transcription. Another interpretation is that colocalization of 2 puncta leads to a $1+1=2$ signal intensity consistent with their data. If there is an increase in transcription, I would expect an even higher intensity level than 2-fold.

We realized that our explanation for original **Fig. 3M, N** in the previous version of our manuscript was confusing. What we suggested was that the observed ratio of intensity/allele between paired vs. unpaired was different from the expected values, as paired vs. unpaired = 2 to 0.6 (observed) instead of 2 to 1 (expected). As exactly pointed out by this reviewer, this may be explained by increased level in paired cases and/or decreased level in unpaired cases. Either cases, we suggest that the difference between observed vs expected is likely reflecting the effect of pairing. Note that we had shown scoring in GSCs and GBs in the original manuscript but we only show scoring in GBs in the revised manuscript for simplicity.

We revised the portion which reads as follows:

“If each *Stat92E* allele contains the same level of nascent transcript, the ratio of intron RNA FISH intensity between a paired and unpaired pattern is expected to be 2:1. However, we found that the average intensity of intron signal per allele was approximately 60% of the expected level (the ratio of paired vs. unpaired loci was approximately 2:0.6, Fig. 4M, N). In contrast, OligoPaint DNA FISH intensity showed 2:1 as expected (Fig. 4O, P). These results supports our hypothesis that physical interaction of homologous regions impacts the levels of nascent transcript, either enhancing transcript between paired alleles and/or suppressing transcript upon unpairing.”

As our *Stat92E* BacTg does not pair with endogenous *Stat92E* locus, we could not use it to plot a standard curve for intron intensities from 0, 1, 2, 3 or 4 copies of *Stat92E* overlapping together. However, we plotted quantification of mRNA dot number for genotypes containing 1, 2, 3, 4 copies of *Stat92E*, which showed linear correlation with copy number and is provided in a new figure (**Supplementary Figure 1H**).

Minor:

1. P-values are missing in Fig. 4O,P
2. High background in Fig. 3I, 5B,H, S2C
3. Spelling in Fig. 4A

We revised these Figures. Thank you for pointing these out.

Reviewer #2 (Remarks to the Author):

In this work, Antel and Inaba use the *Drosophila* male germline stem cell system to study the molecular mechanisms between homologous chromosome pairing, in particular at the Stat92E gene locus, and the transcriptional regulation, mainly the down-regulation from stem cell to more differentiated cells. The results demonstrate that the chromosomal/chromatin changes occur prior to the actual transcript change and likely contributes to the latter change during germline differentiation. These results should lead to new information regarding the developmentally regulated key gene expression during stem cell differentiation and shed light on the downstream effects of asymmetric histone inheritance. This study also provides an example how intrinsic factors act to give rise to an immediate key status change, despite the shallow change of the extrinsic signaling by itself. In this work, the authors also used cutting-edge molecular genetics and cell biology techniques to visualize detailed changes from GSC to its immediate daughter GB cell and so on, therefore provide new tools to study these cells in this model system. Overall, the findings should be of great interest to the readers of Nature Communications. However, there are some concerns and suggestions about the logic flow and data presentation (see below for details when talking about individual parts of the Results). Hope these suggestions will help the authors to make the data presentation more convincing and compelling.

We thank this reviewer for his/her encouraging and important comments. Especially, we appreciate this reviewer's guidance in validating our methodology, which helped us to realize several issues on our original analyses. Below, we categorized this reviewer's concerns and addressed all the comments by performing additional experiments and reanalyzing some data.

1. smFISH signal specificity

1. Figure 1: Stat92E exon smFISH gave out many "dots" everywhere, including nucleus, cytoplasm and extracellular regions. In particular, abundant signals were detected in the hub region. What is the control to make sure these signals are real?

And how to reconcile these results, that Stat92E transcript is present in hub cells and becomes gradually lost during germline differentiation, with the previous results using the Stat reporters, for which I do not think such a broad expression pattern was observed?

Thank you for this comment. The validation of FISH probe was also suggested by the other reviewer. We performed a smFISH experiment with an induced Stat06346 clone and confirmed the signal disappeared completely in the mutant clones (**Supplementary Figure 1A, B**).

Regarding previous results using Stat reporters, we would like to note that a past study has reported that STAT reporters do not accurately reflect Stat92E expression pattern in testes. In particular, a reporter, Socs36E-PZ1647 (Socs36E-PZ) was shown to be strongly expressed in hub cells and CySCs (see the right panel), although Stat92E function in GSCs has been proven in multiple studies. Similarly, other reporters, including widely utilized reporter, P{10XStat92E-GFP} are typically not detectable for unknown reasons.

2. Methodology of smFISH signal quantification

Figure 1: A technical question: For the quantification of Stat92E mRNA levels in 1F, would using dots/~ middle plane introduce any potential bias as it is not necessary that these signals are evenly distributed? Shouldn't the entire dots per cell be more accurate? Some dots are brighter and could they be overlapping multiple dots? The signal intensity could be more accurate if it is not saturated.

Finally, for this figure, as the comparison among different cell types and stages needs internal control for technical reasons, such as FISH efficiency, it should be noted these Ns in 1D, 1F and 1H come from how many individual testes and from how many independent experiments.

Figure 3-5: The same technical question: For the quantification of Stat92E mRNA levels in 3B, H, L, and 4H, L, as well as 5G, would using dots/~ middle plane introduce any potential bias? Shouldn't the entire dots per cell be more accurate? Some dots are brighter and could they be overlapping multiple dots? The signal intensity could be more accurate if it is not saturated.

Thank you for raising these critical points. We realized that we did not explain why we use this single-plane measurement in the previous version of our manuscript.

As shown in above responses, *Stat92E* mRNA is present in CySCs, which are tightly associated with germ cells. Therefore, it was extremely difficult to determine to which cells mRNA dots belong at the surface of cells (an example of a cell surface view is in a new figure, **Fig. 1G**). After trial of several different methods, we found that manual counting at a single plane gives the most reproducible results across different samples.

To ensure our measurements are accurately reflecting the change in mRNA levels in cells, we performed additional smFISH using a probe for a germline-specific gene, *nanos*, which is

known to be downregulated similarly to the *Stat92E* pattern during GSC differentiation, and is not expressed in neighboring CySCs. *nanos* smFISH indeed showed a rapid decrease of dot frequency with progression through developmental stages. Notably, the whole cell measurement and the single plane measurement showed almost identical SIs (Silencing indices; the ratio of mRNA levels between GSC and 2-4 SG). Thus, we concluded that the SI obtained by our method is faithfully reflecting the *Stat92E* downregulation rate. We added a new supplementary figure (**Supplementary Figure 1**) to explain this point.

As pointed out by this reviewer, we agree that some of dots are brighter and could reflect multiple overlapping dots. However, such cases were rare (~1% for *Stat92E*, ~5% for *nanos*) and we therefore do not feel this frequency would significantly bias the total quantification through undercounting, even though we counted these brighter dots also as "1" as we did the less bright dots. We added the description in the Method section as follows:

"We observed brighter dots that likely are the detection of multiple overlapping mRNA molecules. Because such cases were rare (~1% for *Stat92E*, ~5% for *nanos*), we considered the frequency to have negligible effect on the total number and therefore these dots were also counted as "1"."

For FISH efficiency, we used CySC's intron signal as an internal control (The intensity of intron FISH/allele is plotted after divided by CySC's signal) and we used the ratios (GSC/2-4SG) to compare mRNA dot number among different genotypes. We included the information of testes number and biological replicates in all figure legends. All images used for scoring are available in the BioStudies database (<https://www.ebi.ac.uk/biostudies>) under accession number S-BSST829.

3. Effect of cell cycle on transcription and pairing

Another technical question: Given that GSCs have elongated G2 phase, the signals in GSCs should reflect signals from duplicated chromosomes, but how about CySCs and other staged germ cells? In another word, how to avoid effect by different cell cycle stages among different cell types (i.e. germline vs. cyst) and different stages (i.e. stem cell vs. more differentiated cells)? In the figure legend of 1D, it is mentioned that "Cells with more than three puncta were omitted from scoring (see description about Fig 1H)", but I cannot find such a description for Fig. 1H. Do the authors mean Fig. S1B?

Figure 2: The same technical question applies here: For direct comparison of FISH signals, how to avoid effect by different cell cycle stages among different cell types (i.e. germline vs. cyst) and different stages (i.e. stem cell vs. more differentiated cells)? Also, it should be noted the Ns in 2C, F, I, L, O, particularly in 2C, come from how many individual testes and from how many independent experiments.

We appreciate for these insightful comments. First, we apologize that the Fig. 1H was mistakenly cited in the figure legend and we meant to cite **Fig. S1B** (now **Fig. 1F** in the revised manuscript).

i) Effect of cell cycle on pairing

We do see the concern raised by this reviewer such that pairing is regulated in a cell cycle dependent manner and the detected difference of pairing states might be reflecting the difference of cell-cycle stages.

We conducted an additional experiment to clarify these points. We used the FlyFucci line to distinguish S-phase and G2-phase cells, and it revealed that the pairing of *Stat92E* loci still exhibited the same pattern as original analyses (when there was a mix of cell cycle stages) when only G2-phase cells were scored (see **new Fig. 3**), indicating that the observed change of frequency of *Stat92E* pairing between the GSC and the GB stage is unlikely caused by different cell cycle fractions in these stages. The result section for new Fig. 3 reads as follows:

“The change of pairing states does not reflect the difference of cell-cycle stages.

It is thought that somatic homolog pairing may be regulated cell-cycle dependently [5, 48]. Therefore, we wondered if the observed *Stat92E* pairing change during differentiation is developmentally regulated or rather simply reflects the changing fractions of cells in different cell cycle stages. To distinguish these possibilities, we examined the pairing states of *Stat92E* between GSCs and differentiating cells when both populations were in the same cell cycle stage. To this end, we expressed Fly-Fucci, a fluorescent cell cycle marker [49], under the control of the germline driver *nosGal4*, and examined *Stat92E* pairing states by visualizing nascent transcript foci using intron RNA FISH (**Fig. 3A**). A previous study has demonstrated that early germ cells in the *Drosophila* testis lack G1 phase and most germ cells are in G2 or S-phase [35]. We noticed that *Stat92E* transcription became low, often undetectable during S-phase (**Fig. 3A-C**). Therefore, we determined the distance between homologous *Stat92E* loci in G2-phase cells. The distribution of distances between *Stat92E* puncta measured specifically in G2-phase cells did not have a significant difference from the measurement of entire cell populations (**Fig. 3C**), indicating that the observed change of frequency of *Stat92E* pairing between the GSC and the GB stage is unlikely caused by different cell cycle fractions in these stages.”

ii) Effect of cell cycle on nascent transcript

We do see the concern raised by this reviewer about the effect of cell cycle on nascent transcript level, not only because of the difference of copy number of the locus, but also because of the possibility that nascent transcript level might be regulated in a cell-cycle dependent manner.

Indeed, our new analysis using Fly-Fucci line showed that S-phase cells have a significantly lower level of nascent transcript that was often non-detectable (see **new Fig. 3**). Therefore, consideration of cell cycle stage in scoring of nascent transcript is important.

Fig. 4N

GSC/GB pairs normally finish S-phase before cytokinesis. Because we selected cells finished cytokinesis in **Fig. 4N** (**Fig. 3N** in our original manuscript), scored cells should all be in G2 phase. We did not have explanation of this point in the previous version of our manuscript. We added a description of cell-cycle stage of scored GBs in the revised result section for this figure, as follows:

“To assess the direct effect of pairing change on downregulation of transcription, we next attempted to examine levels of nascent transcript between paired and unpaired fractions of cells within the same stage. We specifically focused on the GB stage when the unpaired populations are undergoing the switch from a paired state. The pair of interconnected GSC and GB has been known to synchronously enter S-phase when they remain continuous prior to cytokinesis (**Fig. 4L**) [35]. At this phase, *Stat92E* nascent transcript was almost undetectable (**Fig. 3A**). After completion of cytokinesis, GSCs and GBs are in G2 phase and contain already duplicated chromosomes (4n, **Fig. 4L**). To avoid effects of the cell cycle on transcription, we selected G2-phase GBs, identified as cells located one-cell layer away from the hub and no longer connected to GSCs, and compared their levels of nascent transcript between paired and unpaired alleles.

If each *Stat92E* allele contains the same level of nascent transcript, the ratio of intron RNA FISH intensity between a paired and unpaired pattern is expected to be 2:1. However, we found that the average intensity of intron signal per allele was approximately 60% of the expected level (the ratio of paired vs. unpaired loci was approximately 2:0.6, **Fig. 4M, N**). In contrast, OligoPaint DNA FISH intensity showed 2:1 as expected (**Fig. 4O, P**). These results supports our hypothesis that physical interaction of homologous regions impacts the levels of nascent transcript, either enhancing transcript between paired alleles and/or suppressing transcript upon unpairing.”

Note that we had shown scoring in GSCs and GBs in the original manuscript but we only show scoring in GBs in the revised manuscript for simplicity.

Fig. 1K and Supplementary Figure 4

We realized that the scoring of nascent transcript during the stage progression shown in **Fig. 1K and Supplementary Figure 4** (**Fig. 1H** and **Fig. S3** in the original manuscript) contains S-phase cells with low levels of nascent transcript throughout of stages independent of pairing states. Because maximum levels of nascent transcript (likely from G2 phase cells) still show gradual decrease during stage progression, we consider that our original conclusion, “*Stat92E* nascent transcript is gradually downregulated”, will stay the same. However, we realized that the silencing index (SI) may not be comparable to mRNA scoring. We removed SI from nascent transcript scoring and added notes in figure legends.

We also included the scoring of CySC populations in which *Stat92E* alleles were consistently paired (see **new Supplementary Figure 3D, E**).

4. Interpretation of results regarding to the potential role of pairing on transcription

Also, if the “paired” situation correlates with higher intron smFISH signals compared to “unpaired”, why does the “unpaired” situation in Bac Tg/Df show higher levels of signals in 3H and why do the nos>slmb RNAi (“unpaired”) show higher levels of signals than those in the nos>mrg15 RNAi (“paired”)? However, the “paired” situation in nos>H3T3A (paired) show higher levels of signals than that in nos>H3 (unpaired), consistent with the paired status with higher levels. Or, does it only matter for the change between “unpaired” and “paired”?

And if it does, the mRNA levels do not seem to significantly change between wild-type GSC (mostly paired) and GB (mostly unpaired), as shown in Fig. 1F and 1H? I am trying to get the logic straightened here for these results, but it seems something is missing in this chain.

Thank you for your comment on this important point. As pointed out by this reviewer, disruption of either *Stat92E* pairing or unpairing did not change the initial level of *Stat92E* mRNA in GSCs, indicating that pairing condition is not simply activating transcription and unpairing is not simply repressing transcription. Therefore, we propose that the “change” is what is important. When the *Stat92E* locus needs to become repressive, it requires the change of pairing state, as evidenced by our data showing prompt reduction of *Stat92E* expression is consistently lost in all genotypes with a pairing defect. We have revised the discussion section, as follows:

“Disruption of either *Stat92E* pairing or unpairing did not change the level of *Stat92E* mRNA in GSCs, indicating that pairing condition is not simply activating transcription and unpairing is not simply repressing transcription (Fig. 4E, K, 6G). However, when the *Stat92E* locus needs to become repressive, it requires the change of pairing state, as evidenced by our data showing prompt reduction of *Stat92E* expression is consistently lost in all genotypes with a pairing defect. The mechanism through which the *Stat92E* pairing change facilitates the downregulation of *Stat92E* expression in the *Drosophila* germline remains to be determined.”

5. Other points

Finally, in 2M-O, in this deficiency line, how is the *Stat92E* mRNA level, does it stay high and if so, does this line have any phenotype by itself?

mRNA levels in the *Stat92E* deficiency line, *Df(3R)BSC516*, stayed unchanged (originally shown in old **Fig. 3B** and now in **new Supplementary Figure 1H**).

The phenotypic consequence of *Stat92E* pairing defect was also asked by the other reviewer. We have examined *Stat92E* protein levels in the genotypes in which pairing was perturbed, and found no differences in protein expression pattern compared with the controls. This is consistent with our idea that *Stat92E* protein is also regulated post-transcriptionally

(Original **Fig. S1A** and now in **new Supplementary Figure 6A, B**). Although, we selected *Stat92E* as a model to study pairing effect on transcription, we speculate that other genes, which may be similarly regulated through pairing change, may contribute to the differentiation of GSCs. Revised result section reads as follows:

“Finally, we asked whether the observed *Stat92E* pairing defect has any impact on Stat92E protein distribution. In wild type testes, Stat92E protein level shows a clear reduction in the GB and is almost non-detectable in the SG stage (Supplementary Figure 6A [25, 60]), even earlier than transcription shuts off (Supplementary Figure 6B), indicating that the level of Stat92E is also regulated post-transcriptionally. We examined the pattern of Stat92E protein distribution in genotypes in which *Stat92E* pairing was perturbed. As expected, in all genotypes, Stat92E protein showed normal downregulation as seen in the wild type condition, with high expression in GSCs and immediate reduction in differentiating germ cells (Supplementary Figure 6C-J). It has been shown that the defect in asymmetric histone inheritance disturbs germline differentiation [21]. Therefore, we speculate that other genes that contribute to the differentiation of GSCs may be similarly regulated through pairing change.”

Also, in Fig. 4N, does the GSC show a “paired” signal? If so, why does the cartoon show an “unpaired” scenario?

Thank you for pointing this out, the image was mislabeled, we have revised it accordingly.

I wonder whether the histone status affect the Stat92E enhancer region, can this be tested using the nos>H3T3A with the STAT06346 allele? This could also be investigated using the reporters with the putative Stat-binding sites, given that Stat92E is self-regulated through a positive feedback. This would be very informative if this regulation is through Stat-binding sites, therefore paired situation could enhance the co-activation of both alleles through dimerization.

Thank you for these interesting suggestions. Indeed, the putative enhancer element disrupted in *STAT06346* possesses Stat92E binding sites. Therefore, it is possible that Stat92E itself binds to its enhancer element and promoting pairing.

Stat92E protein is also regulated post-transcriptionally and is downregulated even quicker than the level of mRNA (**new Supplementary Figure 6**). If Stat92E protein is regulating its own transcription through a previously suggested positive feedback mechanism, there is the possibility that the pairing change occurs in response to Stat92E protein reduction.

Gain- or loss- of function analyses of Stat92E protein, such as preventing the degradation of Stat92E protein during ACD or clonal analysis of *Stat92E* point-mutation (for example, *stat92E*^{85C9}) may answer this question. Since the mechanism of Stat92E degradation is completely unknown, we consider this is beyond the scope of current manuscript.

Thank you also for your interesting suggestion of usage of *Stat92E* reporter as a minimum construct of pairing target. This is incredibly interesting idea. However, as we stated in our earlier response, *Stat92E* reporters are not accurately reporting *Stat92E* expression in germline. To pursue this direction, we first need to define minimum elements that can be recognized in the germline, and this would be a most exciting future experiment.

RNAi conditions for Fig. 3 and 5 need to be described in the Methods, for example, is it a constitutively knockdown or temperature-controlled conditional knockdown? This could be an issue to consider if prolonged knockdown leads to cellular defects.

We understand this reviewer's concern regarding indirect effects caused by longer-term knock-down. In prior version of our manuscript, most of RNAi data were based on constitutive knock-down. We have redone all RNAi experiments using temperature-sensitive driver (*nosGal4 delta VP16* combined with *Tub-Gal80ts*) and replaced the data in the revised manuscript. All genotypes showed similar results as reported in the previous version of the manuscript. We also added a temperature shift protocol in the Method section and figure legends.

Another technical question about measuring the distance between intron probe signals or between DNA FISH signals, how to measure their distances on the Z-axis?

The other reviewer raised a similar concern. We realized that the method in which we measured between two puncta was not explained in the previous version of the manuscript.

First, we took series of z-stacks from the testis tip with optimized interval (0.5µm to 1µm step). Then, we selected germ cells in each developmental stage based on their relative location to the hub and connection of cells. We judged whether each cell has a "paired" or "unpaired" pattern. For "paired" cases in which punctae appear as a single spot, we plotted "zero" on each violin plot. When two signals were located in different z-stacks, we calculated the distance by simple math: (actual distance)²= (distance on a plane)²+ z², using excel and plotted the value on each violin plot. Revised Method section reads as follows:

"To measure distances between DNA FISH or RNA intron FISH puncta, we imaged z-stack of entire testis tip with optimized interval (0.5µm to 1µm step) by using a Zeiss LSM800 airy scan with a 63× oil immersion objective (NA=1.4). Using ImageJ, we judged whether a cell has single punctate "paired" or two separate puncta "unpaired" pattern. For all "paired" cases in which punctate appears as a single spot, we plotted "zero" on each violin plot. When two signals were located in different z-stacks, we calculated the distance using an equation,

$$\text{Distance} = \sqrt{x^2 + z^2}$$

where D is distance between punctae, x is a measured 2D distance on a single plane and z is z-distance by Microsoft Excel and plotted values on each violin plot."

To validate that the method we used was accurately reflecting the distance between 2 alleles, we further performed a 3D measurement using Imaris software which offers tools for measurement simply by manually clicking two punctae in different plane of 3D-stack. We confirmed that there is no significant difference between our original measurements and Imaris measurements (new **Supplementary Figure 2**).

This question is a bit remote which is not necessary to be addressed in this work but some discussion would be helpful: What do the authors think about the de-differentiated GSCs? Will they regain the pairing status or not? Another point that the authors could discuss more is whether these changes affect Stat92E's function as a transcription factor (i.e. protein level). For example, does failure in downregulate Stat92E transcript properly lead to tendency of perdurance of Stat92E protein? Finally, a preprint reported less condensed chromosomes in GB compared to GSC (<https://doi.org/10.1101/2021.03.08.434490>), could this difference contribute to the change of pairing of homologous chromosomes from GSC to GB immediately after stem cell asymmetric division? Again, this could be beyond this paper but could be discussed and for future studies.

We have added these points in discussion as suggested (see below). Thank you for bringing this intriguing report to our attention.

“Sister chromatids, with each containing either old or new histones H3 and H4, are inherited to the GSC or GB, respectively. These sister chromatids are hypothesized to have distinct epigenetic information for subsequent cell fate determination [20]. Perturbation of asymmetric histone H3 inheritance results in differentiation defects [21, 61], which suggests that the preferential incorporation of new histones in the GB may be the mechanism actively erasing pre-existing epigenetic memory. Our data suggest that the observed change of *Stat92E* pairing is under the control of biased segregation of sister chromatids, suggesting an interesting possibility that the pairing may transduce inherent epigenetic information to actual gene expression states. The mechanism in which asymmetric inheritance of histones influences pairing states is unknown. Intriguingly, a recent study suggested that the GSC and GB enter S-phase with distinct timing [62]. As a result, two sister chromatids inherit different levels of chromatin condensation, which is required for distinct cell fate decision [62]. This suggests that distinct timing of nucleosome assembly may determine the global epigenetic landscape differently in the GSC and the GB, and there is the possibility that the pairing states of *Stat92E* may be regulated by this process. Future studies determining the dynamic assembly of pairing/anti-pairing factors during S-phase may help to understand the mechanism. Moreover, it would be interesting to assess whether dedifferentiated GSCs, which are GSCs that have returned to the niche after differentiating into GBs or SGs, still retain the correct pattern of pairing/unpairing.”

Minor point: Is reference 30 being mis-referred as it is a paper in *Drosophila* for the developmental roles of JAK-STAT but not on mammalian cells?

We have revised this citation accordingly.

Minor point: In Discussion-- "The Stat92E pairing pattern was unlikely directly affected by the chromosome segregation defect caused by H3T3A expression as the mechanism of homolog pairing and sister chromatid cohesion seem to be distinct in Drosophila cells [59]." This sentence is a bit confusing, the H3T3A affects biased histone inheritance (i.e. sister chromatids carrying old vs. new H3/H4 are mis-inherited) but not chromosome segregation defect which often leads to aneuploidy, etc. Better to clarify this.

We agree that this part was confusing and may not provide a relevant discussion point, so we have removed the portion in the revised manuscript.

REVIEWERS' COMMENTS

Reviewer #1 (Remarks to the Author):

The manuscript has been improved and I have no further concerns.

Reviewer #2 (Remarks to the Author):

The authors have done a satisfactory job in addressing previous questions and comments raised for the initial submission. Therefore, I am enthusiastically supporting acceptance of this work.

The only minor comment is that #62 reference needs an update.